# NAT10-mediated acetylation of NIK mRNA in B cells promotes IgA production

Wan-Jun Jiang[1,5], Xin-Tao Mao[2,5], Wen-Ping Li[2,5], Nicole Jin[2], Yu Wang[3], Guiping Guan [ID][1✉], Jin Jin [ID][2,3✉] & Yi-Yuan Li [ID][2,4✉]

## Abstract

The regulation of IgA expression is crucial for maintaining mucosal immune homeostasis, providing a vital defense mechanism against pathogens at mucosal surfaces. However, the intricate mechanisms governing IgA class-switch recombination and its dysregulation in diseases such as inflammatory bowel disease remain a significant challenge in the field. Our study delves into the significance of IgA regulation in mucosal immunity, focusing on the $N^4$-acetylcytidine (ac$^4$C) in NIK mRNA by NAT10 in B cells. We discovered that NAT10-mediated ac$^4$C stabilizes NIK mRNA, thereby promoting IgA production, which is pivotal for immune defense. Our findings in a B-cell conditional NAT10 knockout mouse model highlight a reduction in IgA expression and a dampened noncanonical NF-κB pathway, suggesting NAT10 as a potential therapeutic target for IgA-related disorders. This research provides novel insights into the post-transcriptional regulation of IgA and underscores the role of NAT10 in modulating mucosal immunity.

Keywords IgA Regulation; NAT10; Noncanonical NF-κB Pathway; B Cell; mRNA Stability
Subject Categories Immunology; RNA Biology; Signal Transduction

## Introduction

The molecular regulation of IgA expression in B cells is a complex and highly regulated process that has been the subject of extensive research in recent years (Lycke and Bemark, 2017). IgA, particularly in its dimeric form, plays a critical role in mucosal immunity, providing a first line of defense against pathogens at mucosal surfaces (Mantis et al, 2011). IgA is the predominant immunoglobulin at mucosal surfaces, where it performs several critical functions (Li et al, 2020). It is produced by plasma cells that have undergone a process of differentiation and class switch recombination (CSR) from other immunoglobulin classes, most commonly IgM (Cerutti, 2008; Stavnezer and Schrader, 2014). The primary function of IgA is to prevent the colonization and invasion of pathogens by binding to their surface antigens, a process known as immune exclusion (Li et al, 2020). This binding can also neutralize toxins and viruses, promote the agglutination and opsonization of pathogens, and facilitate their clearance by immune cells (Blutt et al, 2012). In the gut, IgA plays a crucial role in maintaining the balance between the host and its microbiota (Mantis et al, 2011). It can selectively bind to commensal bacteria, promoting their colonization while excluding pathogens. This process is known as symbiotic IgA, which is thought to be induced by the interaction of the microbiota with the host immune system. Additionally, IgA can modulate the immune response by inhibiting the activation of antigen-specific T cells (Cong et al, 2009), thus preventing unnecessary inflammation.

Class switch recombination (CSR) is a process by which B cells change the class of antibody they produce (Stavnezer et al, 2008). For IgA production, this involves the switch from IgM or IgG to IgA (Cerutti, 2008; Stavnezer and Schrader, 2014). The molecular mechanisms underlying CSR are complex and involve multiple factors. T follicular helper (Tfh) cells are essential for IgA CSR (Kato et al, 2014; Zhang et al, 2020). They provide help to B cells through the interaction of CD40 ligand (CD40L) on Tfh cells with CD40 on B cells (Kato et al, 2014). This interaction, along with cytokines such as IL-4 and TGF-β produced by Tfh cells, promotes the expression of AID and the activation of CSR to IgA (Cazac and Roes, 2000; Sonoda et al, 1989). Cytokines play a critical role in the regulation of IgA CSR. IL-4 and TGF-β are well-known inducers of IgA CSR. IL-6, while typically associated with IgG production, can also influence IgA CSR under certain conditions (Ramsay et al, 1994). The balance and concentration of these cytokines, as well as their timing, are crucial for the efficiency of CSR. Regarding transcription factors, the expression of transcription factors such as Bcl-6, IRF4, and BATF is crucial for the differentiation of B cells into IgA-producing plasma cells (Bunker et al, 2015; He et al, 2016; Liu et al, 2020). These factors are involved in the regulation of gene expression, including the IgA locus. Epigenetic modifications, including DNA methylation and histone modifications, play a significant role in the regulation of IgA CSR (Casali et al, 2021; Hayashi et al, 2020; Sugino et al, 2021). These modifications can

[1]College of Bioscience and Biotechnology, Hunan Agricultural University, 410128 Changsha, China. [2]Center for Neuroimmunology and Health Longevity, the Third Affiliated Hospital of Sun Yat-sen University, 510630 Guangzhou, China. [3]The MOE Key Laboratory of Biosystems Homeostasis & Protection, Zhejiang Provincial Key Laboratory for Cancer Molecular Cell Biology, Life Sciences Institute, Zhejiang University, 310058 Hangzhou, Zhejiang, China. [4]Affiliated Zhuhai People's Hospital, School of Medical Engineering, Beijing Institute of Technology, 519088 Zhuhai, Guangdong Province, China. [5]These authors contributed equally: Wan-Jun Jiang, Xin-Tao Mao, Wen-Ping Li. ✉E-mail: guanguiping@hunau.edu.cn; jjin4@zju.edu.cn; 103200067@seu.edu.cn

affect the accessibility of the IgA locus and the expression of IgA. Furthermore, the discovery of the role of the aryl hydrocarbon receptor (AhR) in the regulation of IgA production has opened new avenues for understanding how environmental factors influence mucosal immunity (Metten, 2015). AhR, a transcription factor that senses environmental cues, has been shown to promote IgA CSR and the differentiation of B cells into IgA-producing plasma cells. Additionally, the role of microRNAs (miRNAs) in the regulation of IgA expression has been explored. Specific miRNAs, such as miR-181a and miR-146a, have been shown to regulate the expression of genes involved in IgA CSR and plasma cell differentiation (Bockmeyer et al, 2016; Tripathy et al, 2023). These findings suggest that miRNAs may serve as potential therapeutic targets for modulating mucosal immunity.

The noncanonical NF-κB pathway has emerged as a critical regulator of IgA production (Jin et al, 2012; Sun, 2017b). This pathway is distinct from the canonical NF-κB pathway, which is typically associated with pro-inflammatory responses (Liu et al, 2017). The non-canonical NF-κB pathway is activated by the interaction of lymphotoxin with its receptor LTβR on B cells (Sun, 2011, 2017a). This interaction leads to the activation of NIK, which in turn activates the IKKα complex (Sun, 2011). The activation of this complex results in the processing and nuclear translocation of the p52 subunit of NF-κB, which is essential for IgA production. The p52/RelB complex formed in the non-canonical NF-κB pathway binds to specific DNA sequences, regulating the expression of genes involved in IgA production. This complex is particularly important for the expression of genes associated with plasma cell differentiation and IgA production (Sun, 2017a). The non-canonical NF-κB pathway also interacts with T cells, particularly Tfh cells, to enhance the production of cytokines that promote IgA CSR and plasma cell differentiation (Hu et al, 2011). This crosstalk is essential for the coordinated regulation of IgA production in the context of mucosal immunity. Despite considerable advances, the intricate regulatory mechanisms governing noncanonical NF-κB pathways remain enigmatic.

In this study, we discovered that B cells, upon stimulation with anti-CD40 and BAFF, upregulate the acetyltransferase NAT10. NAT10 selectively acetylates the mRNA of the key kinase *Map3k14* (NIK), thereby promoting its stability and protein translation. Mice with B cell-specific NAT10 deficiency exhibit a significant reduction in IgA expression and suppression of noncanonical NF-κB pathway activation. Our research elucidates a novel regulatory mechanism of noncanonical NF-κB and identifies a core factor for IgA production in vivo. Our findings also suggest that targeted modulation of NAT10 may play a crucial role in the treatment of IgA nephropathy or inflammatory bowel disease (IBD).

# Results

## NAT10 is highly expressed in IgA⁺ B cells of patients with IBD

The patients with IBD often exhibit a decrease in IgA levels, which may impair the mucosal barrier function and contribute to the dysregulation of the gut microbiota (Bamias et al, 2023; Leake, 2014). To validate the levels of IgA in the serum and feces of patients with IBD, we collected clinical samples from 30 healthy volunteers and 30 patients with newly diagnosed IBD (Fig. EV1A). Our findings revealed a significant reduction in serum IgA levels in IBD patients compared to healthy controls (Figs. 1A and EV1B). Additionally, the proportion of IgA-expressing B cells in colonic tissue was markedly decreased in the IBD group (Fig. 1B). These reductions in IgA levels may compromise mucosal immune responses, potentially exacerbating inflammation and increasing susceptibility to infections in IBD patients. The acetylation of cytidine at position 4 (ac⁴C) within the coding regions of genes enhances the process of translation, while ac⁴C modifications in the 5' untranslated regions (UTRs) influence the synthesis of proteins at the initiation stage. Interestingly, we found that IgA⁺ B cells exhibited higher ac4C modification levels (Fig. 1C).

NAT10, also known as N-acetyltransferase 10, functions as an enzyme that catalyzes the ac⁴C modification, a post-transcriptional modification known to enhance mRNA stability and translation efficiency (Xie et al, 2023). NAT10 plays a crucial role in various cellular processes, including the regulation of gene expression and the maintenance of genomic stability (Yan et al, 2023). Consistent with the increased ac⁴C levels, the protein abundance of NAT10 is markedly upregulated in IgA⁺ B cells (Fig. 1D). To elucidate the relationship between NAT10 expression and B cell activation, total B cells were stimulated with recombinant B-cell activating factor (BAFF). The results demonstrated a rapid increase in both mRNA and protein levels of NAT10 in activated B cells (Fig. 1E,F). We further observed a significant reduction in NAT10 expression in colonic B cells from IBD patients, consistent across both mRNA and protein levels (Fig. 1G,H). We further performed flow cytometry-assisted isolation of lamina propria-derived IgA⁺ B cells from matched cohorts. qPCR analyses demonstrated a significant downregulation of NAT10 transcripts in IBD-derived IgA⁺ B cells compared to healthy controls (Fig. EV1C). These findings corroborated the model wherein NAT10 deficiency disrupted IgA⁺ B cell survival or differentiation. Moreover, in patients with Crohn's disease (CD), NAT10 expression in colonic B cells exhibited a strong negative correlation with disease severity index (Fig. 1I). These findings suggest that NAT10 may play a role in the regulation of IgA production by B lymphocytes in IBD.

## NAT10 is dispensable for B cell development and maturation

To investigate the impact of a specific genetic manipulation on B cell function, we generated a mouse model with conditional deletion of NAT10 in B cells (NAT10ᶜᴷᴼ) using a Cre-loxP system. Mice carrying loxP-flanked *Nat10* alleles were crossed with mice expressing Cre recombinase under the control of the B cell-specific *Cd19* promoter (*Cd19*-Cre) (Fig. EV2A). The knockout efficiency and specificity of NAT10 were verified at the protein level (Fig. EV2B). Unlike mice with a global deletion of NAT10 (NAT10⁻/⁻ mice) or T cell-conditional NAT10 KO mice, which exhibit developmental abnormalities, the NAT10ᶜᴷᴼ mice were born at the expected Mendelian ratios and displayed no gross abnormalities in growth or survival.

In the spleen, B cells are categorized into distinct subsets, each with unique functions in the immune response (Allman and Pillai, 2008). These subsets include the follicular (FO) B cells, marginal zone (MZ) B cells, transitional B cells (T1 and T2), and follicular

mantle (FM) B cells. FO B cells are primarily involved in humoral immunity, MZ B cells respond quickly to blood-borne pathogens, T1 and T2 B cells are immature and maturing forms, respectively, and FM B cells surround the germinal centers, contributing to the maintenance of B cell homeostasis. The B cell-specific deletion of NAT10 did not appear to affect B cell development or maturation, as evidenced by the presence of comparable B cell subpopulations in the spleen (Fig. 2A,B). Germinal center (GC) B cells are crucial for the generation of high-affinity antibodies and the formation of memory B cells through a process that involves somatic hypermutation and class-switch recombination (Klein and Dalla-Favera, 2008). The absence of NAT10 also did not affect the proportion of GC B cells (Fig. 2C), as it might be more involved in cellular differentiation and reprogramming rather than directly influencing B cell subset proportions. Furthermore, NAT10 deficiency did not contribute to the development and maturation

of B cell subsets in bone marrow (BM) (Fig. 2D). Loss of NAT10 in B cells also did not affect germinal center formation, as suggested by the normal structure of dark and light zone (Fig. 2E). Finally, we found that loss of NAT10 in B cells did not affect tissue inflammation (Fig. EV2C).

## NAT10 is essential for IgA production in B cell

To study the role of NAT10 in the activation and maturation of B cells, we performed vaccinations on mice with normal genetics and NAT10cKO. The vaccines contained the hapten NP (4-hydroxy-3-nitrophenylacetyl) linked to two different carriers: keyhole limpet hemocyanin (NP-KLH), and the hydrophilic polysaccharide Ficoll (NP-Ficoll). These were selected to mimic T cell-dependent, and T cell-independent type 2 immune responses (Fig. 3A). Our findings indicated that the levels of IgA specific to the antigen were

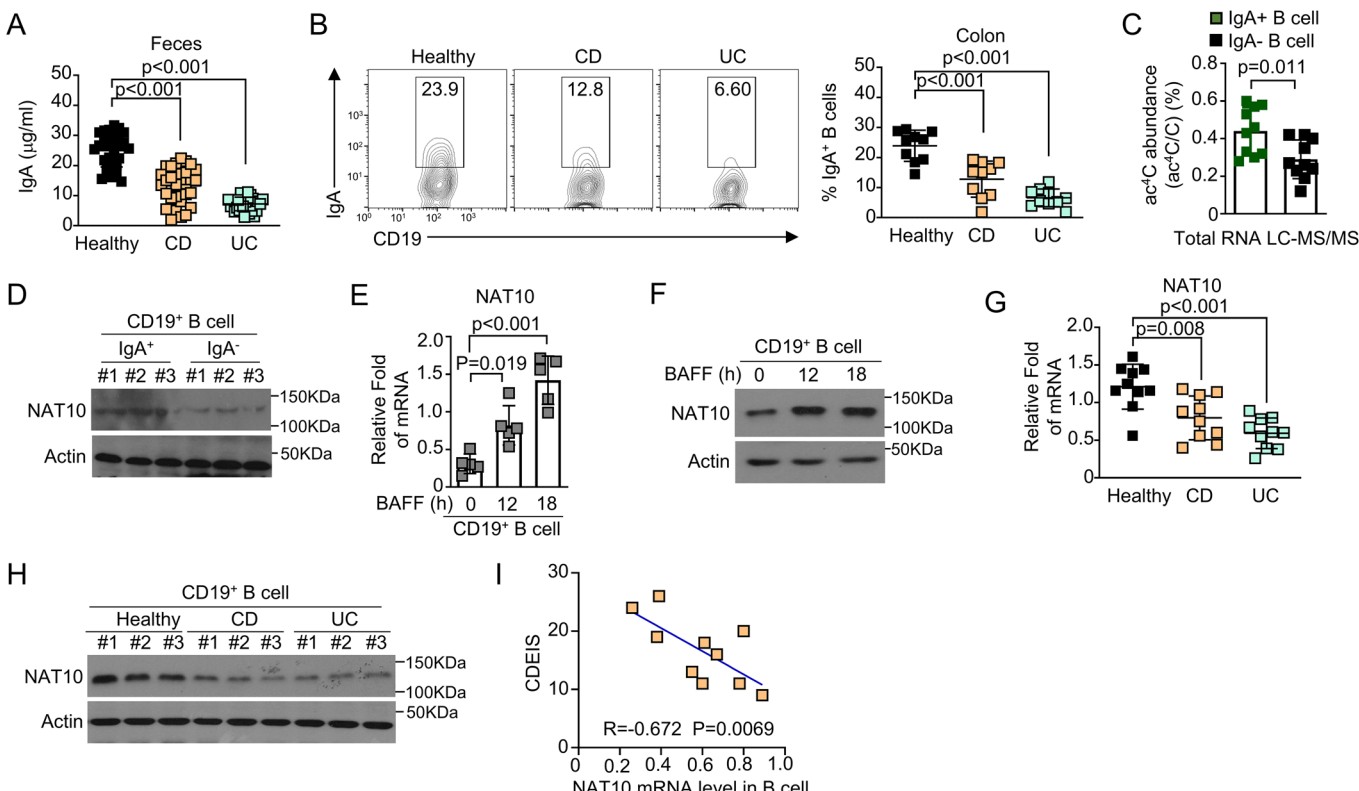

**Figure 1. NAT10-mediated regulation of IgA in B cells and its association with IBD pathogenesis.**

(A) ELISA was used to quantify IgA levels in fecal samples from healthy donors and newly diagnosed patients with Crohn's disease (CD) or ulcerative colitis (UC) ($n = 30$). (B) The proportion of IgA+ B cells within the total CD19+ B cell population from colonic tissues of healthy donors and newly diagnosed CD or UC patients ($n = 10$) was assessed using flow cytometry. The percentages of cells in each gated region are indicated. (C) LC–MS/MS was employed to determine the relative ac4C/C content in total RNA extracted from IgA+ and IgA- B cells from colonic tissues of healthy volunteers ($n = 10$). (D) Immunoblotting (IB) was used to measure NAT10 protein levels in IgA+ and IgA- B cells from the colonic tissues of healthy volunteers ($n = 3$). (E) CD19+ B cells isolated from healthy volunteers were stimulated with BAFF (200 ng/ml) for the indicated times, and *Nat10* mRNA levels were analyzed by RT-qPCR ($n = 5$). The data are presented as fold changes relative to ACTB mRNA levels, normalized using Bio-Rad CFX Manager 3.1. (F) Following BAFF stimulation for the indicated times, total cellular extracts from CD19+ B cells isolated from healthy volunteers were subjected to immunoblotting to detect NAT10 protein. (G) NAT10 expression was analyzed by qPCR in total CD19+ B cells isolated from the colonic tissues of healthy donors and newly diagnosed CD or UC patients ($n = 10$). Results are shown as fold changes relative to *Actb* mRNA levels, normalized with Bio-Rad CFX Manager 3.1. (H) NAT10 protein levels were assessed by immunoblotting in total CD19+ B cells from colonic tissues of healthy donors and newly diagnosed CD or UC patients ($n = 3$). (I) A scatterplot showing the linear regression correlation between NAT10 mRNA levels in B cells from the colonic tissues of newly diagnosed CD patients ($n = 10$) and their disease activity index (CDAI). All data are representative of biological replicates in three independent experiments. Data are represented as the means ± SDs. The significance of differences (C) was determined by t test, and those (A, B, E, G) were determined using one-way ANOVA with Newman–Keuls post-hoc test. **$P < 0.01$; ***$P < 0.005$. Source data are available online for this figure.

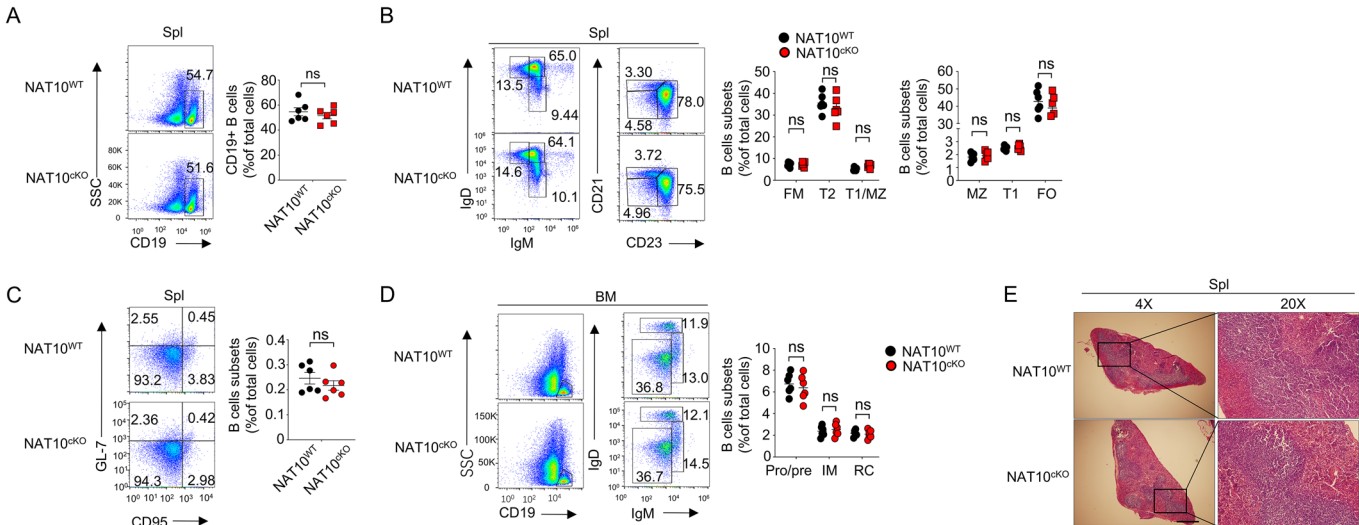

**Figure 2. Generation and characterization of B cell-specific NAT10-deficient mice for immune function analysis.**

(A–C) Flow cytometry was used to assess the percentages (numbers within quadrants) of different B cell subsets within the CD19+ splenic cells in the spleens of 6-week-old WT and NAT10cKO mice (n = 6). FM, follicular mature (IgMintIgDhigh); FO, follicular (CD21intCD23high); T1, transitional 1 (IgMhighIgDlow, CD21lowCD23low); T2, transitional 2 (IgMhighIgDhigh); MZ, marginal zone (IgMhighIgDlow, CD21highCD23low). (D) Flow cytometry was also employed to analyze the proportions of recirculating (RC), immature (IM), and early developmental (ProPre) B cell stages (n = 6). (E) Tissue sections of spleens from 6-week-old WT and NAT10cKO mice were performed with hematoxylin and eosin (H&E) staining (n = 3). Scale bar = 10 μm. All data are representative of biological replicates at least three independent experiments. Data are represented as the means ± SDs. The significance of differences (A–D) was determined by t test. ns no significance. Source data are available online for this figure.

significantly reduced in the NAT10cKO mice, whereas the IgM and IgG levels remained largely unchanged (Fig. 3B). In the non-immunized state, the levels of IgM and IgG in the serum were found to be similar between NAT10cKO mice and their wild-type counterparts of the same age, throughout the study period (Fig. 3C). Notably, the NAT10cKO mice had a significant decrease in the production of antigen-specific IgA compared to the controls (Fig. 3D). By the age of 12 months, the NAT10cKO mice also displayed a downregulation of IgA production without affecting total concentration of IgM and IgG (Fig. EV3A). Additionally, NAT10cKO mice had showed no difference in serum autoantibody levels against nuclear antigens and double-stranded DNA (Fig. EV3B). IgA is predominantly found in the mucosal lining of the gastrointestinal tract, where it plays a crucial role in immune defense by neutralizing pathogens and preventing their adherence to the intestinal epithelium. Consistently, NAT10cKO mice exhibited significantly reduced IgA deposition in the colon compared to the control (Fig. 3E). Consequently, the presence of NAT10 in B cells is essential for controlling systemic and colonic IgA levels.

## NAT10 in B cell is essential for mucosal homeostatic and inflammatory condition

In order to explore the role of NAT10 in modulating IgA-associated inflammatory conditions, we employed a DSS-induced acute colitis model to mimic the clinical progression of IBD. WT and NAT10cKO mice were exposed to 3% DSS for a continuous period of five days to assess their vulnerability. We evaluated the impact by tracking changes in body weight, calculating the Disease Activity Index (DAI), and observing survival rates. The data in Fig. 4A–C illustrated that the NAT10cKO mice exhibited more weight reduction, a higher DAI, and a lower survival rate post-DSS exposure compared to the WT mice, with

only subtle variations noted in the H2O-treated cohorts. Macroscopic analyses revealed that the colons of NAT10cKO mice were considerably shortened under DSS-induced conditions compared to those of the WT mice, as depicted in Fig. 4D. Concurrently, microscopic examination revealed heightened inflammation and increased damage to the mucosal epithelium in NAT10cKO mice, marked by increased leukocyte infiltration, as shown in Fig. 4E. Moreover, flow cytometric analysis revealed a significant increase in the presence of colon-infiltrating macrophages, identified by CD11b+F4/80+ markers, in NAT10cKO mice in response to DSS (Fig. 4F). Intestinal IgA plays a critical role in maintaining the equilibrium of the gut microbiota, as it helps to form a protective barrier that prevents the translocation of potentially harmful bacteria while preserving the balance of beneficial microbes, thus contributing to the overall health of the gut ecosystem. Consistently, the NAT10cKO mice exhibited a markedly elevated bacterial count in their fecal matter (Fig. 4G), coinciding with a diminished capacity of antibodies in the bloodstream to bind bacteria (Fig. 4H). Unlike the wild-type controls, the NAT10cKO mice exhibited a decrease in secretory IgA (sIgA) concentrations within their fecal matter and serum (Fig. 4I), coupled with a reduced presence of IgA+ B cells during colitis (Fig. 4J).

IgA is the first line of defense against inhaled antigens and allergens (Mantis et al, 2011). IgA antibodies are particularly effective at neutralizing viruses and bacteria, and they help maintain the integrity of the mucosal barrier, which is essential for preventing respiratory infections. In order to ascertain the impact of NAT10-regulated IgA on viral infections, we conducted a study assessing the immune response to viral challenge in NAT10cKO mice. Both WT and NAT10cKO mice were administered the H7N9 virus intranasally, and their condition was tracked for a week. NAT10cKO mice, when compared to their WT counterparts, experienced significant weight loss and an increased incidence of

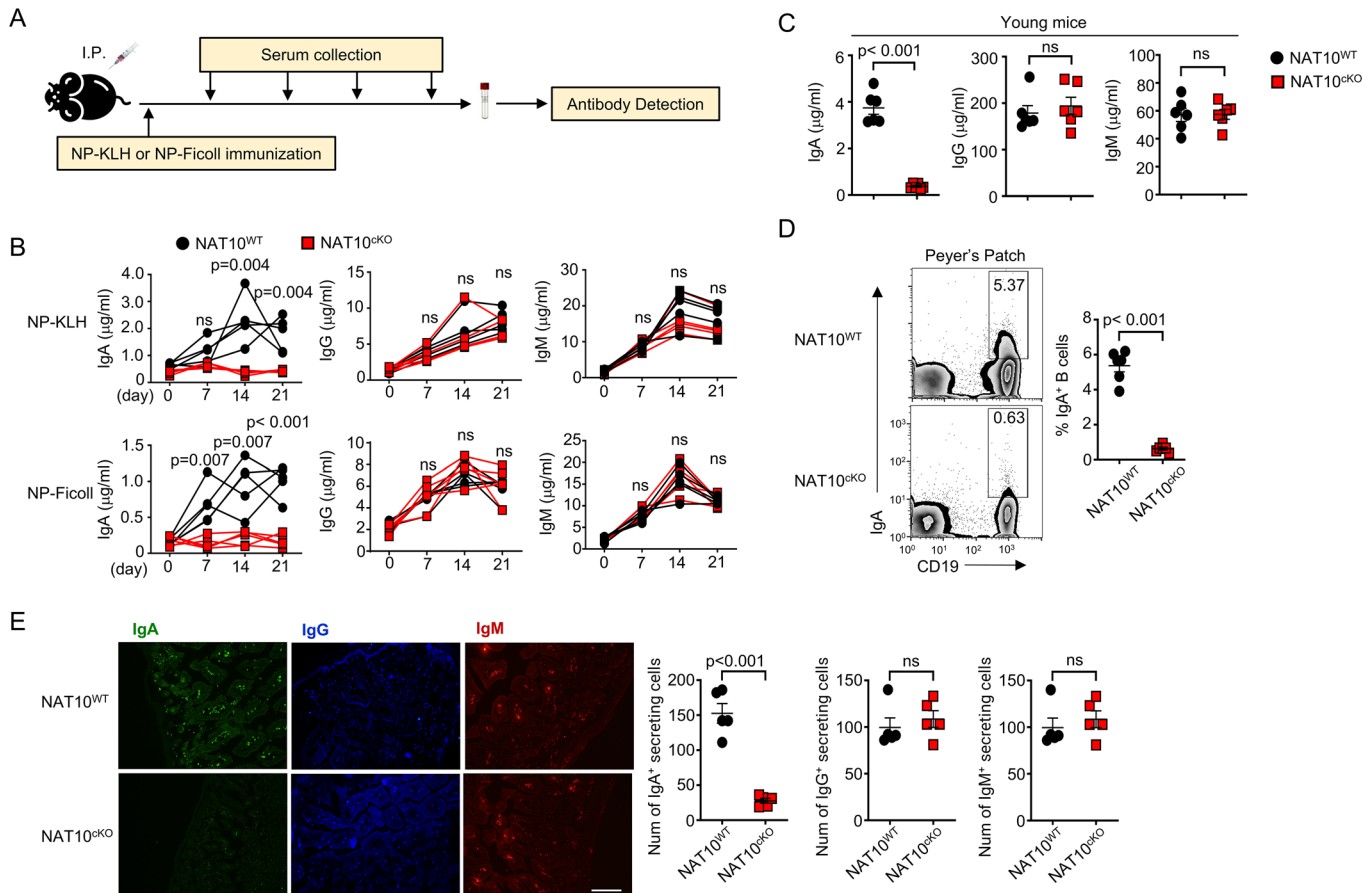

**Figure 3. Impact of NAT10 on B cell activation and IgA production in mice.**

(A) Diagram illustrating the experimental design. (B) Quantification of NP-specific antibody subclasses in the serum of 6-week-old WT and NAT10cKO mice, following intraperitoneal immunization with NP-KLH or NP-Ficoll, measured by enzyme-linked immunosorbent assay (ELISA) ($n = 5$). (C) Total antibody subclass levels in the serum of 6-week-old WT and NAT10cKO mice immunized with sheep red blood cells (SRBCs), determined by ELISA ($n = 6$). (D) Analysis of the proportion of IgA+ B cells within the spleens of 12-month-old WT and NAT10cKO mice immunized with SRBCs, evaluated 7 days post-immunization by FACS and subsequent quantification ($n = 6$). (E) Immunofluorescence microscopy of IgA deposition in colon tissue sections from 12-month-old WT and NAT10cKO mice, with a subsequent quantification of immunoglobulin deposits ($n = 5$). Scale bar = 10 μm. All data are representative of biological replicates at least three independent experiments. Data are represented as the means ± SDs. The significance of differences (B–E) was determined by $t$ test. **$P < 0.01$; ***$P < 0.005$. Source data are available online for this figure.

illness (Fig. EV3C,D). Subsequent analysis of the lung tissue from the infected mice revealed a notably higher quantity of viral copies in NAT10cKO mice (Fig. EV3E). A lower sIgA was detected in bronchoalveolar lavage (BAL) of NAT10cKO mice 7 days post H7N9 challenge (Fig. EV3F). Furthermore, H&E staining revealed that the NAT10cKO mice had developed acute and severe viral pneumonia, characterized by intense immune cell infiltration and considerable damage to the lung tissue (Fig. EV3G). Collectively, these findings suggest that the absence of NAT10 in B cells heightened the susceptibility of mice to H7N9 virus infection, as evidenced by enhanced viral propagation and inflammation induced by the influenza A virus in the early stages of infection.

## NAT10 promotes noncanonical NF-κB-mediated IgA induction

We next evaluated the function of NAT10 in the modulation of B cell activation and maturation. Our findings showed that B cells

lacking NAT10 exhibited decreased proliferation rates compared to their WT counterparts when activated by non-canonical NF-κB stimulators, such as anti-CD40, or BAFF (Fig. 5A). Conversely, this heightened proliferation was not detected in NAT10-deficient B cells activated by canonical NF-κB activators (Fig. 5A). It is important to highlight that the absence of NAT10 did not significantly change the IgA class-switching driven by TGF-β alone, yet it significantly impaired the induction of IgA+ B cells (Fig. 5B) and the production of sIgA (Fig. 5C) in response to anti-CD40 and BAFF. The initiation of immunoglobulin class-switch recombination is marked by the emergence of germline transcripts (GLTs) and is contingent upon the action of the cytidine deaminase AID. Our data revealed that the absence of NAT10 downregulated the expressions of genes encoding α-GLT and AID under anti-CD40 and BAFF stimulation (Fig. 5D). In summary, these results underscore the pivotal function of NAT10 in the regulation of class-switching to IgA, thereby elucidating the irregular IgA synthesis observed in NAT10cKO mice.

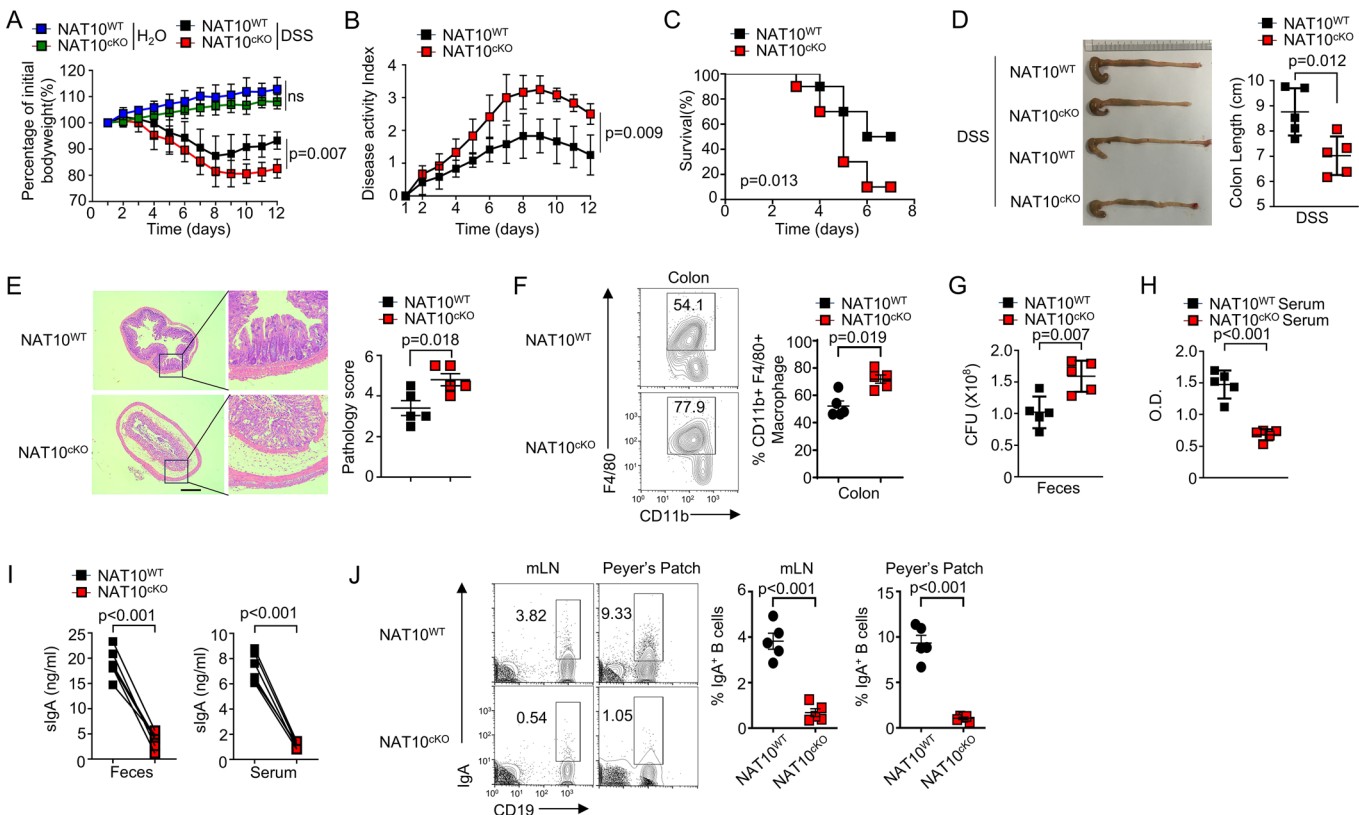

**Figure 4. NAT10 deficiency in B cells exacerbates inflammation in murine models of IBD.**

(A–C) Weight loss (A) ($n = 6$), disease activity score (B) ($n = 6$), and survival rate (C) ($n = 10$) in DSS-treated WT and NAT10cKO mice. (D, E) Measurements of colon length (D) ($n = 5$), and histopathological analysis using hematoxylin and eosin (H&E) staining (E) ($n = 5$) on day 12 following DSS treatment in WT and NAT10cKO mice. Scale bar = 200 μm. (F) Flow cytometry (FACS) analysis showing macrophage (CD11b+F4/80+) populations in representative plots from multiple mice ($n = 5$). (G) Bacterial colony-forming units (CFU) counted in the feces of WT and NAT10cKO mice on day 12 of colitis ($n = 5$). (H) Optical density (O.D.) values reflecting the amount of bacteria bound by serum antibodies in WT and NAT10cKO mice on day 12 post-DSS treatment ($n = 5$). (I) Serum and mucus-rich fluid supernatants were collected, the latter transferred into 10 mL of phosphate-buffered saline (PBS), and secretory IgA (sIgA) levels in the colon were measured via ELISA ($n = 6$). (J) The percentage of IgA+ B cells among the B cell population in the mediastinal lymph nodes (mLN) and Peyer's patches of WT and NAT10cKO mice on day 12 post-DSS treatment were measured by FACS, along with the quantification of these results ($n = 5$). All data are representative of biological replicates at least three independent experiments. Data are represented as the means ± SDs. The significance of differences (B–J) was determined by $t$ test, and those (A) were determined using two-way ANOVA. **$P < 0.01$; ***$P < 0.005$. Source data are available online for this figure.

To assess the impact of Remodelin, an inhibitor of NAT10 acetylation, on B cell proliferation and IgA production, Remodelin was applied to inhibit the acetyltransferase activity of NAT10, thereby disrupting the ac4C and influencing processes like mRNA stability and nuclear architecture (Dalhat et al, 2021). As expected, B cells treated with Remodelin demonstrated diminished proliferation and survival when stimulated with anti-CD40 or BAFF in vitro (Fig. EV4A) and were less readily differentiated into IgA+ B cells (Fig. EV4B). Additionally, the expression of GLTs and AID and IgA level was significantly reduced in the presence of Remodelin (Fig. EV4C,D). Together, these findings highlight a pivotal role for NAT10 in the regulation of IgA production induced by noncanonical NF-κB activators.

## NAT10 positively regulates the activation of noncanonical NF-κB pathway

We next examined the role of NAT10 in regulating signal transduction during the activation of B cells. When exposed to

the canonical NF-κB activators anti-IgM, NAT10cKO B cells did not enhance the activation of the canonical NF-κB proteins p50 or RelA (Fig. EV5A). The absence of NAT10 also exhibited no difference in MAP kinases or AKT activation (Fig. EV5B,C). We then investigated NAT10's involvement in NF-κB activation through noncanonical NF-κB stimulators anti-CD40 and BAFF. The activation of nuclear p52 and RelB by anti-CD40 and BAFF was significantly reduced in NAT10cKO B cells compared to WT B cells (Fig. 6A,B). This reduction correlated with an increased accumulation of the p52 precursor, p100, in the cytoplasm of NAT10cKO B cells, indicating that NAT10 was required for promoting p100 processing (Fig. 6A,B). These observations were consistent with the notable impact of NAT10 deficiency on IgA induction by anti-CD40 and BAFF, suggesting a specific function for NAT10 in the regulation of noncanonical NF-κB pathways. TRAF3, a key regulator of the noncanonical NF-κB pathway, normally inhibits NIK by facilitating its degradation. Stimulation with anti-CD40 and BAFF induced TRAF3 degradation, resulting in NIK accumulation and sustained NF-κB activation. Interestingly, we observed that

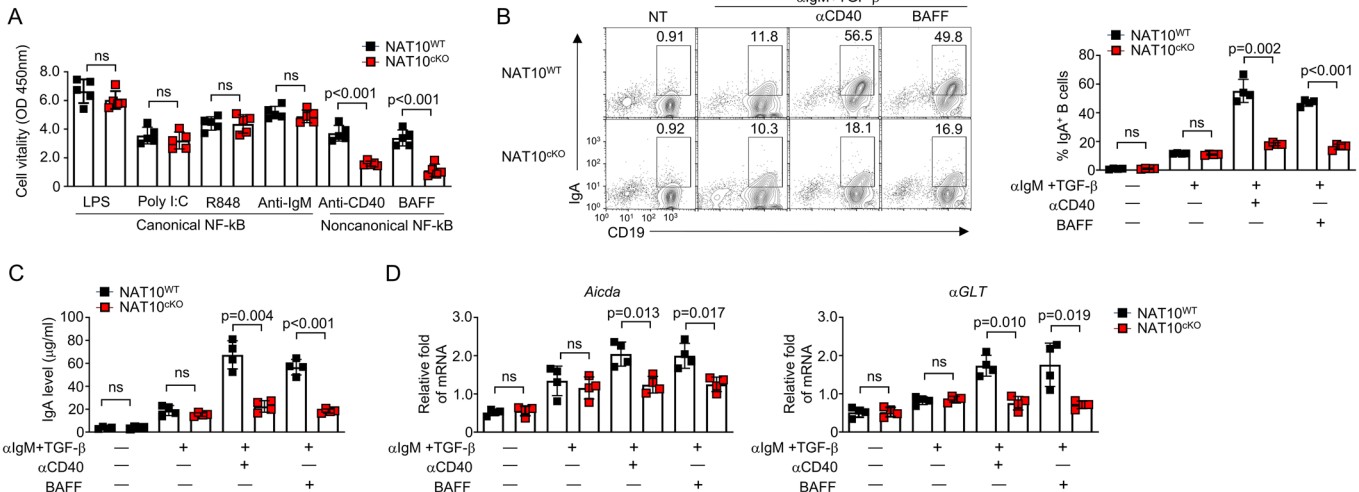

**Figure 5. NAT10 promotes B cell activation, and IgA production through noncanonical NF-κB signaling.**

(A) Proliferation assays of splenic B cells were conducted in vitro, either without stimulation (NT) or with the specified inducers. These assays measured the proliferation of splenic B cells in response to the indicated stimuli ($n = 5$). (B) Intracellular IgA levels in splenic B cells were analyzed after 5 days of culture with αIgM (10 µg/ml) alone or combined with TGF-β (2 ng/ml), anti-CD40 (α-CD40; 1 µg/ml), or BAFF (200 ng/ml). The percentage of IgA+ B cells was quantified by flow cytometry, with numbers shown in the outlined regions ($n = 4$). (C) ELISA was used to measure IgA levels in the supernatants of splenic B cells cultured for 5 days with αIgM alone or together with TGF-β, anti-CD40, or BAFF ($n = 4$). (D) qPCR was performed to assess α-GLT and AID mRNA expression in splenic B cells after 5 days of culture under the same conditions as in (B). The results are presented as fold changes relative to *Actb* mRNA levels and normalized using Bio-Rad CFX Manager 3.1 ($n = 4$). All data are representative of biological replicates at least three independent experiments. Data are represented as the means ± SDs. The significance of differences (A) was determined by $t$ test, and those (B–D) were determined using one-way ANOVA with Newman–Keuls post-hoc test. **$P < 0.01$; ***$P < 0.005$. Source data are available online for this figure.

NAT10 deficiency hindered NIK induction in response to both anti-CD40 and BAFF (Fig. 6C,D). The reduced levels of NIK protein in NAT10cKO cells were not attributable to upstream signaling disruptions, as TRAF3 degradation remained unaffected by the absence of NAT10 (Fig. 6C,D). Previous study demonstrated that BAFF promoted TBK1 phosphorylation and induces NIK phosphorylation, thereby destabilizing NIK. To evaluate whether NAT10 influences this axis, we compared TBK1 phosphorylation levels in BAFF-stimulated WT and NAT10 KO B cells. Our data revealed no significant difference in TBK1 activation between these groups (Fig. EV5D), thus demonstrating that NAT10-mediated regulation of the non-canonical NF-κB pathway operates independently of TBK1 activity.

The NAT10 mutant G641E has been shown to significantly impact the acetyltransferase activity of NAT10 (Dalhat et al, 2021). This mutation can lead to a disruption in the enzyme's ability to acetylate its substrates, which may have downstream effects on various cellular processes. In B lymphocytes harboring a targeted disruption of NAT10, we ectopically expressed the WT NAT10 (NAT10-WT) or the NAT10-G641E mutant allele. Utilizing the mCherry tag intrinsic to these constructs, we successfully isolated the reconstituted B cells via fluorescence-activated cell sorting (FACS) (Fig. EV5E). Upon stimulation with BAFF, the reconstituted B cells expressing NAT10-WT exhibited a marked enhancement in proliferative capacity, a response that was notably absent in the B cells expressing the NAT10-G641E mutant (Fig. 6E). Analogous findings were corroborated during the differentiation of IgA+ B cells (Fig. 6F). Within this cohort of reconstituted B cells, the presence of NAT10-WT significantly augmented the levels of NIK protein, an effect that was not replicated by the NAT10-G641E

mutant (Fig. 6G), suggesting a diminished functional capacity of the mutant allele in modulating NIK protein expression.

## NAT10 stabilizes the mRNA of NIK via promoting its ac⁴C modification

NAT10 acetylates RNA by transferring an acetyl group to the N4 position of cytidine bases (Cai et al, 2017). This acetylation can enhance mRNA stability by protecting it from degradation, thereby influencing the availability of mRNA for translation and subsequent protein synthesis (Wang et al, 2022). The ac⁴C modification by NAT10 has been implicated in the regulation of gene expression, including the stabilization of specific mRNAs such as PAN RNA in Kaposi's sarcoma-associated herpesvirus (KSHV), which is crucial for viral reactivation and replication. Given that the absence of NAT10 results in decreased activation of the non-canonical NF-κB pathway, we first examined the mRNA levels of key components, including *Traf3*, *Nfkb2*, *Map3k14* and *RelB*, in this pathway. qPCR analysis revealed that NAT10-deficient B cells exhibited reduced *Map3k14* (NIK) mRNA expression following stimulation with either anti-CD40 or BAFF, while the mRNA levels of *Traf3*, *Nfkb2*, and *RelB* remained comparable to those in WT controls (Fig. 7A). Additionally, in HEK293T cells, we observed that NAT10 enhanced both the mRNA and protein levels of overexpressed NIK, whereas the expression of the control gene *Gfp* showed no significant difference (Fig. 7B,C). These findings suggest that NAT10 specifically upregulates *Map3k14* mRNA expression and its corresponding protein levels. Cycloheximide (CHX) is utilized in mRNA stability studies to inhibit general protein synthesis, allowing researchers to track the decay rate of specific

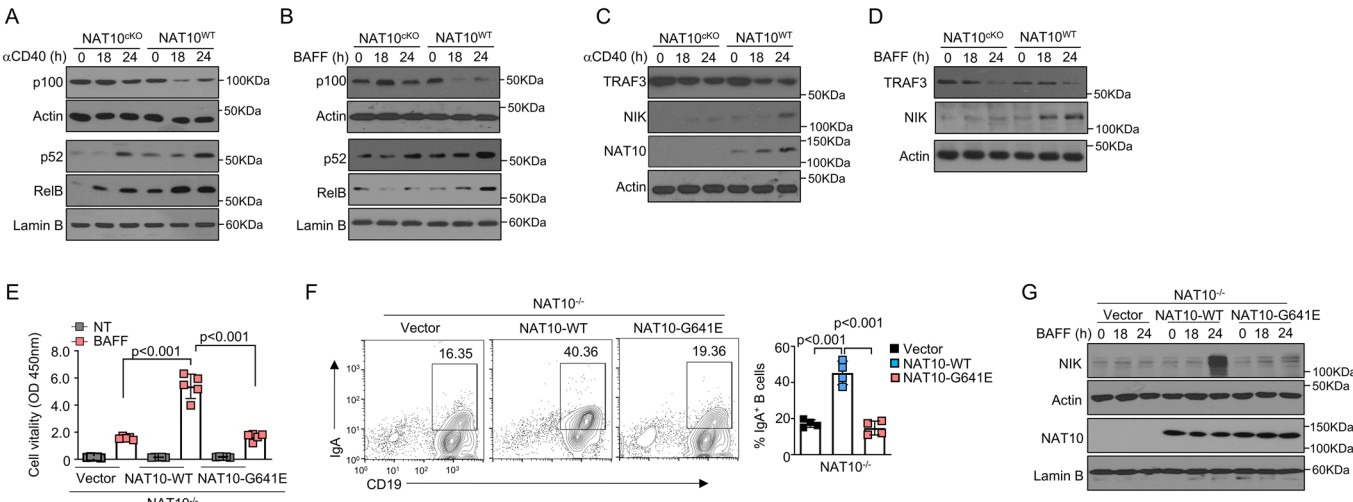

**Figure 6. NAT10 modulates noncanonical NF-κB signaling through promoting the accumulation of NIK.**

(A, B) IB analysis of noncanonical NF-κB components in cytoplasmic (CE) and nuclear (NE) extracts from NAT10-deficient and WT B cells, stimulated with anti-CD40 (α-CD40; 1 μg/ml) or BAFF (200 ng/ml). (C, D) IB analysis using total cell extracts from WT and NAT10-deficient B cells, stimulated as indicated. (E) Immunoassay of NAT10$^{-/-}$ B cells stimulated with LPS (5 μg/ml) for 8 h, followed by infection with a retroviral vector expressing GFP and either WT NAT10 (NAT10-WT) or the NAT10-G641E mutant. Infected cells (GFP$^+$) were sorted via flow cytometry and stimulated with BAFF, after which in vitro proliferation assays were performed ($n = 5$). (F) Proportion of IgA$^+$ cells among purified NAT10$^{-/-}$ B cells overexpressing the indicated genes, cultured with various combinations of IgA inducers, and subjected to in vitro IgA class switching. Analysis was performed using flow cytometry, based on intracellular IgA and GFP staining ($n = 4$). (G) IB analysis of whole-cell lysates from NAT10$^{-/-}$ B cells overexpressing the indicated genes as described in (E). All data are representative of biological replicates at least three independent experiments. Data are represented as the means ± SDs. The significance of differences (E, F) was determined using one-way ANOVA with Newman–Keuls post-hoc test. **$P < 0.01$; ***$P < 0.005$. Source data are available online for this figure.

mRNAs in the absence of new transcription, thereby assessing their stability. By measuring the decrease in mRNA levels over time in the presence of CHX, NAT10-deficient in B cell displayed a faster reduction compared to their littermates (Fig. 7D). The RIP-qPCR assay, in particular, is utilized to validate known RNA targets associated with a specific RNA-binding protein (RBP). The RIP-qPCR assay has been instrumental in demonstrating that NAT10 can specifically bind to the mRNA of *Map3K14* (Fig. 7E). This interaction is crucial as it modulates the stability of *Map3K14* mRNA, thereby influencing its availability for translation and downstream cellular processes.

Using publicly available data (PACES) (Zhao et al, 2019), we predicted the presence of N4-ac⁴C modification sites in NIK mRNA. The results identified conserved ac⁴C modification sites within the C-terminal region of both human and mouse NIK mRNA (Fig. 7F). To further explore this, we generated an ac⁴C mutant of NIK mRNA (NIKac⁴C mut). Following transfection of HEK293T cells overexpressing NAT10, the mRNA and protein levels of WT NIK were significantly higher than those of the NIKac⁴C mut, whereas the control gene *Gfp* confirmed equal transfection efficiency between the two groups (Fig. 7G,H). Consistent with the phenotype observed in NIK-deficient cells, the mRNA stability of the NIKac⁴C mut was markedly reduced (Fig. 7I). Additionally, NIK$^{-/-}$ B cells transfected with either NIKWT or NIKac⁴C mut were evaluated for their ability to differentiate into IgA$^+$ B cells. The NIKac⁴C mut displayed a clear loss of IgA induction capability (Fig. 7J), suggesting that NAT10 stabilizes NIK mRNA via ac⁴C modification, thereby supporting IgA$^+$ B cell differentiation through the activation of the noncanonical NF-κB pathway.

## Discussion

The regulation of IgA production is of paramount importance in mucosal immunity, providing a critical line of defense against pathogens at mucosal surfaces (Blutt et al, 2012). IgA, particularly in its dimeric form, plays a pivotal role in immune exclusion, neutralization of toxins and viruses, and modulation of the immune response to maintain homeostasis within the gut microbiota (Bunker et al, 2015). Our study significantly supplements the field by identifying NAT10 as a novel regulator of IgA production in B cells. Diminished NAT10 expression may underlie the depletion of intestinal IgA$^+$ B cells observed in IBD pathogenesis. While BAFF and CD40L signaling pathways are recognized as collaborative regulators of IgA$^+$ plasma cell maintenance, their functional impairment in IBD likely exacerbates B cell apoptotic susceptibility and compromises mucosal IgA synthesis. Insufficient BAFF/CD40L signaling might reduce NIK mRNA production, creating a negative feedback loop that further diminishes NAT10 activity. Additionally, there are other mechanisms may also regulate NAT10 expression: 1. Inflammatory cytokines: Pro-inflammatory cytokines such as TNF-α, IL-6, or IL-1β, which are elevated in IBD, may suppress NAT10 expression. These cytokines could activate signaling pathways (e.g., NF-κB or STAT3) that directly or indirectly downregulate NAT10 transcription; 2. Microbial influence: The characteristic dysbiosis in IBD could alter microbial metabolites (e.g., short-chain fatty acids) or pathogen-associated molecular patterns (e.g., LPS), which might impair NAT10 expression via Toll-like receptor (TLR) signaling. This discovery expands the current understanding of post-transcriptional regulation in IgA production and offers new insights into the complex interplay

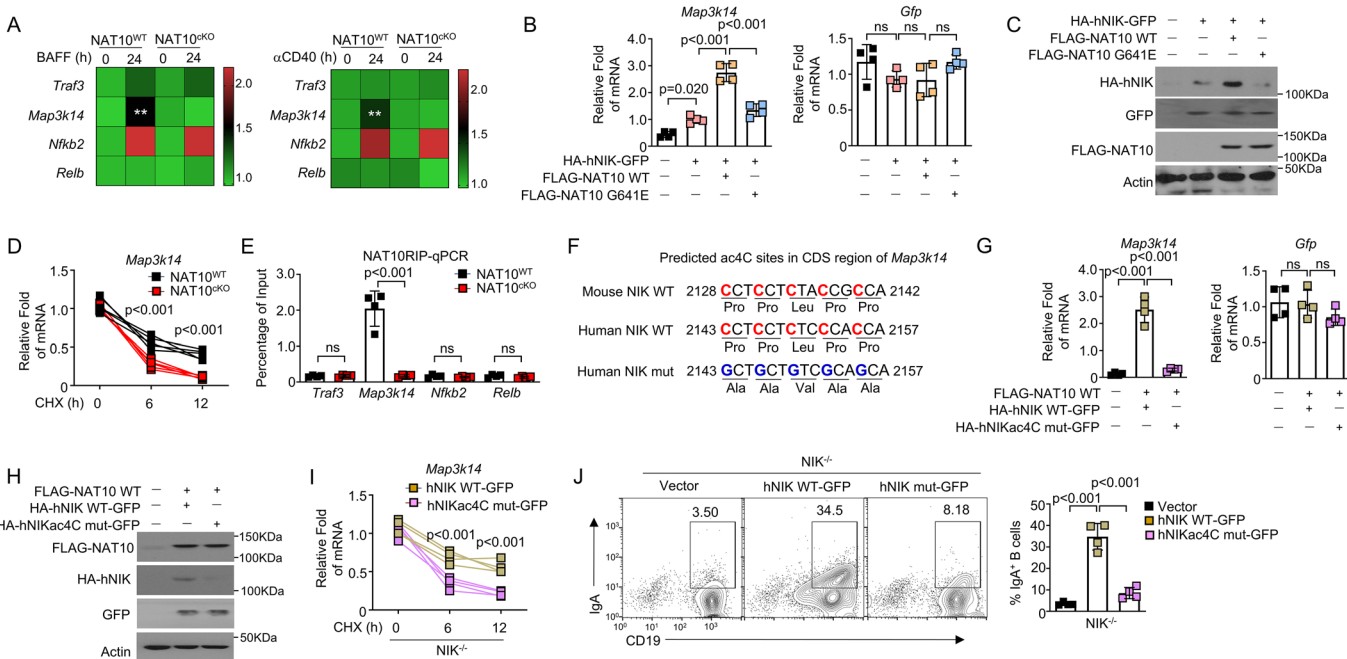

**Figure 7. NAT10-mediated acetylation of NIK mRNA regulates noncanonical NF-κB pathway and IgA+ B cell differentiation.**

(A) qPCR analysis performed on RNA extracted from spleen B cells derived from WT and NAT10cKO mice, stimulated with either anti-CD40 (α-CD40; 1 µg/ml) or BAFF (200 ng/ml) (n = 4). (B) qPCR analysis of *Map3k14* and *Gfp* mRNA in HEK293T cells, transfected with various combinations of expression vectors for hemagglutinin-tagged human NIK (HA-hNIK)-GFP and Flag-tagged NAT10 WT or the NAT10-G641E mutant (n = 4). (C) IB analysis of specific proteins in HEK293T cells transfected with different plasmid constructs. (D) mRNA stability assays in WT and NAT10-deficient B cells pretreated with BAFF (200 ng/ml) for 24 h, followed by the addition of 3 µg/ml CHX to inhibit transcription. Cells were harvested at the indicated times post-CHX treatment, and the levels of NIK mRNA were quantified by qPCR (n = 5). Results are expressed as fold changes relative to *Actb* mRNA, normalized using Bio-Rad CFX Manager 3.1. (E) RNA immunoprecipitation (RIP) was conducted using an anti-NAT10 antibody, followed by qPCR to detect RNA enrichment in NAT10−/− B cells overexpressing either NAT10 WT or NAT10-G641E, and stimulated with BAFF (200 ng/ml) for 24 h (n = 4). (F) Sequence alignment of ac4C modification sites in NIK mRNA targeted by NAT10. Canonical bases are highlighted in red, and mutated ones in blue. (G, H) qPCR (G) and IB (H) analyses of *Map3k14* and *Gfp* mRNA levels in HEK293T cells transfected with various plasmid combinations (n = 4). (I) mRNA stability assays in NIK−/− B cells infected with different plasmids expressing the specified proteins. These B cells were pretreated with BAFF (200 ng/ml) for 24 h before the addition of 3 µg/ml CHX to stop transcription. Cells were harvested at the indicated times post-CHX treatment, and NIK mRNA levels were assessed by qPCR (n = 4). Data are shown as fold changes relative to *Actb* mRNA, normalized using Bio-Rad CFX Manager 3.1. (J) Proportion of IgA+ cells among purified NIK−/− B cells overexpressing the indicated genes, cultured with different combinations of IgA-inducing factors, and analyzed for IgA class switching in vitro by flow cytometry using intracellular IgA and GFP staining (n = 4). All data are representative of biological replicates at least three independent experiments. Data are represented as the means ± SDs. The significance of differences (E) was determined by t test, and those (B, D, G, I, J) were determined using one-way ANOVA with Newman–Keuls post-hoc test. **P < 0.01; ***P < 0.005. Source data are available online for this figure.

between epigenetic modifiers and immune responses. Our findings highlight the potential of NAT10 as a therapeutic target for modulating mucosal immunity, particularly in the context of IgA-related pathologies such as IgA nephropathy and IBD (Bamias et al, 2023; Mantis et al, 2011; Stamellou et al, 2023).

Our study reveals a unique function of NAT10 in B cells that diverges from its characterized roles in other cell types. Contrary to its established function in rRNA acetylation, which is integral to cell cycle regulation and ribosome biogenesis (Xie et al, 2023), NAT10 in B cells does not appear to play a significant role in rRNA modification. This distinction underscores the cell-type-specific functions of NAT10 and suggests that B cells may have evolved alternative mechanisms for rRNA regulation. The differential requirement of NAT10 in rRNA acetylation between B cells and other cells may be attributed to the unique role of B cells in the adaptive immune response. B cells undergo class switch recombination (CSR) and somatic hypermutation, processes that demand precise control over gene expression, which may involve distinct epigenetic modifiers (Jeevan-Raj et al, 2011; Vaidyanathan and

Chaudhuri, 2015). The specific mechanisms underlying this cell-type specificity warrant further investigation and could provide valuable insights into the evolution of gene expression regulation in immune cells.

NAT10 has been reported to regulate a variety of genes and molecules involved in diverse cellular processes, including cell proliferation, apoptosis, and genomic stability (Hu and Lu, 2024; Tan et al, 2018). Our study identifies a novel and specific role for NAT10 in the acetylation and stabilization of NIK mRNA in B cells. This discovery is significant as it links NAT10 to the non-canonical NF-κB pathway, a key regulator of IgA production. The acetylation of NIK mRNA by NAT10 represents a previously uncharacterized mechanism of post-transcriptional regulation in B cells. Our findings suggest that NAT10-mediated acetylation of NIK mRNA enhances its stability, promoting the activation of the non-canonical NF-κB pathway and subsequent IgA production. This specific regulation of NIK by NAT10 in B cells has not been described previously and represents a significant advancement in our understanding of the molecular mechanisms underlying IgA class-switch recombination.

While our study provides novel insights into the role of NAT10 in IgA regulation, it is not without limitations. Firstly, the study primarily focuses on the in vitro and in vivo models, which may not fully recapitulate the complexity of human mucosal immunity. Secondly, the mechanistic understanding of how NAT10 selectively acetylates NIK mRNA in B cells is not fully elucidated. Future studies should aim to dissect the molecular interactions between NAT10 and NIK mRNA, and how these interactions are regulated in response to various immune stimuli. Additionally, the long-term effects of NAT10 modulation on mucosal immunity and its potential as a therapeutic target require further investigation in clinical settings. Despite these limitations, our study lays the groundwork for future research and opens new avenues for exploring the role of RNA acetylation in immune regulation.

In conclusion, our study provides a comprehensive analysis of NAT10's role in IgA production by B cells. We have identified a unique function of NAT10 in the regulation of NIK mRNA stability and have highlighted the potential of NAT10 as a therapeutic target for modulating mucosal immunity. Further research is needed to fully understand the molecular mechanisms underlying NAT10's function in B cells and to explore its potential in the treatment of IgA-related disorders.

# Methods

### Reagents and tools table

| Reagent/resource | Reference or source | Identifier or catalog number |
|---|---|---|
| **Experimental models** | | |
| Nat10 floxed mice | Gemphamatech, Nanjing, China | Cat# T007971 |
| Rag1-/- mice | Gemphamatech, Nanjing, China | Cat# Cat# T004753 |
| B6.129P2(C)-Cd19tm1(cre)Cgn/J mice | Jackson Lab | Cat# 006785 |
| B6N.129-Map3k14tm1Rds/J mice | Jackson Lab | Cat# 025557 |
| IBD patients | Zhejiang University | Collected ourselves |
| **Recombinant DNA** | | |
| NAT10-WT (mouse) | Li et al, 2024 | Construct ourselves |
| NAT10-G641E (mouse) | Li et al, 2024 | Construct ourselves |
| pCLXSN-NIK WT-GFP (Human) | – | Construct ourselves |
| pCLXSN-NIKac4C mut-GFP (Human) | – | Construct ourselves |
| **Antibodies** | | |
| PE-CY7-anti-CD19 | eBioscience | Cat# 25-0193-82 |
| FITC-anti-IgA | BD Biosciences | Cat# 559354 |
| PerCP-anti-IgM | eBioscience | Cat# 46-5790-82 |
| APC-anti-IgD | eBioscience | Cat# 17-5993-82 |
| FITC-anti-CD21 | eBioscience | Cat# 11-0219-42 |
| PE-anti-CD23 | eBioscience | Cat# 12-0232-82 |
| PE-anti-GL-7 | eBioscience | Cat# 12-5902-82 |
| FITC-anti-CD95 | eBioscience | Cat# 11-0959-42 |
| PE-F4/80 | eBioscience | Cat# 12-4801-82 |
| Anti-NAT10 | Proteintech | Cat# 13365-1-AP |

| Reagent/resource | Reference or source | Identifier or catalog number |
|---|---|---|
| Mouse monoclonal anti-β-Actin (clone AC-74) | Sigma | Cat# A2228 |
| Anti-p100/p52 | Abcam | Cat# ab175192 |
| Anti-RelB | Abcam | Cat# ab309084 |
| Anti-Lamin B | Abcam | Cat# ab16048 |
| Anti-TRAF3 | Abcam | Cat# ab239357 |
| Anti-GFP | Abcam | Cat# ab290 |
| **Oligonucleotides and other sequence-based reagents** | **Sequence** | |
| Human AICDA Forward | TGAAGAGGCGTGACAGTGCTAC | N/A |
| Human AICDA Reverse | GAGATGTAGCGGAGGAAGAGCA | N/A |
| Human LMNA Forward | ATGAGGACCAGGTGGAGCAGTA | N/A |
| Human LMNA Reverse | ACCAGGTTGCTGTTCCTCTCAG | N/A |
| Human CCL3 Forward | ACTTTGAGACGAGCAGCCAGTG | N/A |
| Human CCL3 Reverse | TTTCTGGACCCACTCCTCACTG | N/A |
| Human TIMP1 Forward | GGAGAGTGTCTGCGGATACTTC | N/A |
| Human TIMP1 Reverse | GCAGGTAGTGATGTGCAAGAGTC | N/A |
| Human NAT10 Forward | GGATTGCCTCAACATCACTCGG | N/A |
| Human NAT10 Reverse | CGTTGGAGGAAAACTTCAGAGGC | N/A |
| Human ID1 Forward | GTTGGAGCTGAACTCGGAATCC | N/A |
| Human ID1 Reverse | ACACAAGATGCGATCGTCCGCA | N/A |
| Human TNFRSF17 Forward | TCTTTGGCAGTTTTCGTGCTAATG | N/A |
| Human TNFRSF17 Reverse | CCAGGTCAATGTTAGCCATGCC | N/A |
| Human ACTIN Forward | CACCATTGGCAATGAGCGGTTC | N/A |
| Human ACTIN Reverse | AGGTCTTTGCGGATGTCCACGT | N/A |
| Mouse Aicda Forward | GCCACCTTCGCAACAAGTCT | N/A |
| Mouse Aicda Reverse | CCGGGCACAGTCATAGCAC | N/A |
| Mouse aGLT Forward | CAAGAAGGAGAAGGTGATTCAG | N/A |
| Mouse aGLT Reverse | GAGCTGGTGGGAGTGTCAGTG | N/A |
| Mouse Traf3 Forward | CAGCTAGTCTGCGGTGTGAGAA | N/A |
| Mouse Traf3 Reverse | GGCACCTCAGACTTATCTTGGC | N/A |
| Mouse Map3k14 Forward | GGAATACCTCCACTCACGAAGG | N/A |
| Mouse Map3k14 Reverse | CTGTGAGCAAGGACTTTCCCAG | N/A |
| Mouse Nfkb2 Forward | GGCAGACCAGTGTCATTGAGCA | N/A |
| Mouse Nfkb2 Reverse | CAGCAGAAAGCTCACCACACTC | N/A |
| Mouse Relb Forward | TGTGGTGAGGATCTGCTTCCAG | N/A |
| Mouse Relb Reverse | TCGGCAAATCCGCAGCTCTGAT | N/A |
| Mouse Actin Forward | CGTGAAAAGATGACCCAGATCA | N/A |
| Mouse Actin Reverse | CACAGCCTGGATGGCTACGT | N/A |
| Gfp Forward | AGTCCGCCCTGAGCAAAGA | N/A |
| Gfp Reverse | TCCAGCAGGACCATGTGATC | N/A |

| Reagent/resource | Reference or source | Identifier or catalog number |
|---|---|---|
| NAT10 flox genotyping Forward | GGAACCATGAGTATTGTAGCCTGC | N/A |
| NAT10 flox genotyping Reverse | CTATTGGCTGTGACTTCAGCAGAC | N/A |
| CD19-Cre genotyping WT Forward | GTCCAGGTCCCTGACGTCTG | N/A |
| CD19-Cre genotyping WT Reverse | AGAGGGAGGCAATGTTGTGC | N/A |
| CD19-Cre genotyping KI Forward | GACGATGAAGCATGTTTAGCTGG | N/A |
| CD19-Cre genotyping KI Reverse | AGAGGGAGGCAATGTTGTGC | N/A |
| **Chemicals, enzymes and other reagents** | | |
| Recombinant human BAFF | Biolegend | Cat# 559605 |
| NP-KLH | Santa Cruz | Cat# sc-396478 |
| NP-Ficoll | Biosearch | Cat# Cat# F-1420F-10 |
| DSS | MilliporeSigma | Cat# 9011-18-1 |
| Vancomycin | MCE | Cat# HY-B0671 |
| Ampicillin | MCE | Cat# HY-B0522 |
| Neomycin | MCE | Cat# HY-150520 |
| LPS | Sigma | Cat# L2880-10MG |
| Poly I:C | Sigma | Cat# 31852-29-6 |
| R848 | Sigma | Cat# 144875-48-9 |
| Anti-IgM | Jackson ImmunoResearch, | Cat# 115-066-020 |
| Anti-CD40 | Thermo Fisher Scientific | Cat#16-0402-82 |
| Recombinant human TGF-b | Abcam | Cat# Ab50036 |
| Mouse IgA ELISA Kit | Thermo Fisher Scientific | Cat# EMIGAX5 |
| Human IgA ELISA Kit | Thermo Fisher Scientific | Cat# Cat# BMS2096TEN |
| Mouse IgM ELISA Kit | Thermo Fisher Scientific | Cat# 88-50470-88 |
| Mouse IgG ELISA Kit | Thermo Fisher Scientific | Cat# 88-50400-22 |
| **Software** | | |
| ImageJ | Schneider et al, 2012 | https://imagej.nih.gov/ij/ |
| 10X Cell Ranger package | 10X Genomics | https://support.10xgenomics.com |
| FlowJo | Treestar | https://www.flowjo.com |
| Prism | GraphPad | https://www.graphpad.com |
| **Other** | | |

## Human subjects

All studies were performed with the approval of the Medical Ethics Committee of the Third Affiliated Hospital of Sun Yat-sen University (ID: A2023 552 01). Informed written consent was obtained from all patients who participated in the human study. Before data analysis, all patients were diagnosed with CD or UC based on endoscopic, radiologic, and histopathological criteria. Patients without histories of gut inflammation, immune-mediated diseases, cancer, or a family history of intestinal cancer, who came for routine screening, served as controls. During the endoscopic screening, the endoscopic disease activity was evaluated by experienced GI doctors based on the Simple Endoscopic Score for Crohn's disease (SES-CD) and Mayo Endoscopic Score for UC.

Colon biopsies were obtained after ethical approval and informed consent at the Third Affiliated Hospital of Sun Yat-sen University.

## Ethics statement

This study was conducted in strict accordance with all applicable ethical standards and legal regulations. All research involving human participants has been approved by the Third Affiliated Hospital of Sun Yat-sen University, with the approval number A2023 552 01. Prior to any research procedures, all participants were fully informed about the study's objectives, methods, potential risks, and benefits, and provided their written informed consent. Throughout the research process, the privacy and personal information of all participants were safeguarded to ensure the confidentiality and security of the data. Animal experiments were conducted in compliance with the guidelines of the Animal Care and Use Committee of Sun Yat-sen University and were granted approval with the number A20231013. In the course of the experiments, we were committed to minimizing the suffering of animals and took all necessary measures to ensure their welfare.

## Experimental animals

NAT10 floxed ($Nat10^{fl/fl}$) mice were generated using CRISPR-Cas9-mediated knock-in technology by Gemphamatech, Nanjing. To create experimental groups, these Nat10-floxed mice were bred with CD19-Cre transgenic mice from Jackson Laboratory, yielding age-matched $Nat10^{+/+}Cd19^{Cre/+}$ (referred to as WT) and $Nat10^{fl/fl}Cd19^{Cre/+}$ (referred to as Nat10$^{cKO}$) mice. NIK-deficient mice, which were supplied by Amgen, had a 129SvEv genetic background, while the other strains used in the study were of a mixed B6;129 background. All mice were bred and housed in a pathogen-free facility at the Laboratory Animal Center of Sun Yat-sen University. The maintenance of these animals adhered to specific pathogen-free conditions, and all experiments were carried out in compliance with protocols approved by the Institutional Animal Care and Use Committee at the Third Affiliated Hospital of Sun Yat-sen University (ID: A20231013). No animal was excluded in this research. All animal experiments were conducted in accordance with the Reporting checklist for study using laboratory animals.

## Randomisation and confounder control

Mice were randomly allocated to control and treatment groups using a computer-generated random number sequence (block randomisation stratified by litter). Cage positions within the animal facility were systematically rotated every 48 h to minimise location-related environmental bias. All experimental procedures (e.g., injections, sample collection) were performed in randomised order daily.

## Blinding

Group allocation was concealed from experimenters during animal handling, data collection, and outcome assessment (e.g., flow cytometry analysis). The investigator generating the

randomisation list did not participate in experimental procedures or data analysis. Unblinding occurred only during final statistical interpretation.

## Cell, plasmids, antibodies, and reagents

### Cell

The HEK293T cell line was generously provided by Prof. Shao-cong Sun. These cells were maintained in DMEM supplemented with 10% FBS and 1% streptomycin/penicillin.

### Plasmid

The pcDNA expression vectors encoding HA-tagged human NIK were a gift from Prof. Shao-cong Sun. Site-directed mutagenesis using the QuickChange II Site-Directed Mutagenesis Kit (Stratagene) was used to generate pCLXSN (GFP)-based mouse NIKac4C mutants. We cloned both the FLAG-tagged NAT10 and its catalytically inactive mutant (NAT10 G641E) in-house.

### Antibody

Goat anti-mouse IgM F(ab')2 (anti-IgM) and anti-mouse CD40 (553788), used for B cell stimulation, were obtained from Jackson ImmunoResearch Laboratories and BD Bioscience, respectively. NAT10 antibody (13365-1-AP) came from Proteintech, while antibodies against RelB (C-19), IKBα (C-21), p65 (C-20), ERK (K-23), phospho-ERK (E-4), JNK (C-17), p38 (H-147), p105/p50 (C-19), AKT1 (B-1), Lamin B (C-20), NIK (H248), and TRAF3 (C-20) were sourced from Santa Cruz Biotechnology. Additional antibodies like anti-Actin (C-4), anti-HA (12AC5), anti-FLAG (M2), anti-HA HRP (3F10), and anti-FLAG HRP were purchased from Millipore Sigma. Phosphorylation-specific antibodies, including phospho-AKT (Ser473; D9E), phospho-IκBα (Ser32; 9241), phospho-JNK (Thr180/Tyr185; 9251), phospho-p38 (Thr180/Tyr182; 9211), and phospho-p105 (Ser933, 18E6), were obtained from Cell Signaling Technology Inc. Fluorescence-labeled antibody reagents used for flow cytometry and cell sorting are listed in the section "Flow cytometry".

### Reagents

Recombinant human TGF-β and BAFF were from PeproTech. LPS (derived from *Escherichia coli* strain 0127:B8) was acquired from Sigma-Aldrich and Enzo Life Sciences.

## Flow cytometry

Single-cell suspensions from the specified tissues were stained with fluorescently labeled antibodies and analyzed using a Beckman Cytoflex S flow cytometer. The antibodies used for flow cytometry included: PE-CY7 anti-CD19 (1:100, 552854), PerCP-Cy5.5 anti-IgM (1:100, 550881), APC anti-IgD (1:100, 560868), FITC anti-IgA (1:100, 559354), FITC anti-CD21 (1:100, 561769), FITC anti-CD95 (1:100, 556640), PE anti-CD23 (1:100, 561773), PE anti-GL7 (1:100, 561530), PE anti-F4/80 (1:100, 565410), and Alexa-750 anti-CD11b (1:100, 557657). All antibodies were sourced from BD Biosciences.

## Mouse immunization, antibody detection, and renal function analysis

Age-matched WT and NAT10^cKO mice (6–8 weeks old) were immunized via intraperitoneal injection (i.p.) with 0.2 ml of NP-

KLH or NP-Ficoll (0.1 mg/ml in PBS). Blood samples were collected at specific time points post-immunization, and serum was analyzed using ELISA kits from Southern Biotechnology Associates.

For renal function analysis, serum and urine samples were collected from unimmunized mice at 12 months of age. Auto-antibodies against dsDNA and nuclear antigens were measured using specific ELISA kits (Alpha Diagnostic Intl, Inc.). Renal function was assessed by measuring urinary protein levels (Rat Urinary Protein Assay Kit, Chondrex, Inc.), blood urea nitrogen (Urea Nitrogen Reagent Set, Bio-Quant, Inc.), and serum creatinine (Creatinine Assay Kit, Cayman Chemical Company).

## DSS-induced colitis

For lethality analysis, mice were given 3.5% (w/v) DSS (molecular weight 36,000–50,000 Da; MP Biomedicals) in their drinking water for 6 days, followed by 6 days of regular water. Body weight was recorded every other day for up to 8 days. After this period, mice were sacrificed for colon length measurement, histological evaluation, and immune cell isolation from the colonic mucosa. Disease activity was clinically assessed using a validated scoring system. Colonic macrophages were isolated from the colon tissue by FACS sorting, using CD11b and F4/80 markers.

## Colony-forming units (CFU) calculation

Throughout the experiment, SPF facilities, sterile food, and water were strictly maintained. On day 12, fecal samples (20 mg) were collected as described. Each sample was diluted in 1 mL of sterile PBS, thoroughly mixed, and then diluted 10,000-fold. From the diluted bacterial solution, 50 µL was plated onto solid LB agar and spread evenly using a sterile spreader. The plates were inverted and incubated overnight at 37 °C, after which bacterial colonies were counted to calculate CFUs.

## Virus and infection

All in vivo and in vitro infections were conducted using the influenza virus strain A/HangZhou/1/2013 (H7N9, GenBank accession code KC853766), provided by the Center for Disease Control and Prevention of Zhejiang Province. For in vivo studies, 6-week-old mice were intranasally infected with H7N9 virus ($5 \times 10^4$ pfu in 50 µl PBS). Survival rates and weight loss were tracked daily for up to 14 days post-infection (dpi). For pathogenesis studies, mice were euthanized at specific time points post-infection, and their lungs, spleens, and livers were aseptically harvested for further analysis. In vitro infection experiments involved infecting cells with H7N9 virus at a multiplicity of infection (MOI) of 1, as detailed in the figure legends. After 1 h of viral adsorption at 37 °C, the cells were washed with PBS and cultured in DMEM containing 2 µg/ml TPCK-treated trypsin. Viral stocks were propagated in embryonated chicken eggs, and virus titers were determined using the 50% tissue culture infectious dose (TCID$_{50}$) method in MDCK cells.

## Real-time quantitative RT-PCR (qPCR)

Total RNA was extracted using TRIzol reagent (Life Technologies), and cDNA synthesis was performed using RNaseH-MMLV

reverse transcriptase (TAKARA). The resulting cDNA was diluted and subjected to RT-qPCR using a CFX96 Touch machine (Bio-Rad). Gene expression was analyzed using the relative quantification method, with Actb serving as the reference gene. Primer sequences for the qPCR analysis are provided in "Reagents and Tools Table". all qPCR results were normalized against β-actin (Actb) mRNA levels using the ΔΔCt method. Data normalization and statistical analysis were performed utilizing Bio-Rad CFX Manager 3.1 software, which implements automatic baseline correction and cycle threshold determination algorithms. The software's default normalization protocol centers group means at unity (1.0-fold change) to optimize visual interpretation of relative expression differences while maintaining mathematical integrity of the dataset.

## ELISA

To assess IgA levels in serum, feces, and bronchoalveolar lavage (BAL), samples were flushed and homogenized in endotoxin-free PBS. After centrifugation at $800 \times g$ for 15 min, the supernatants were collected and stored at $-80\,°C$ for future analysis. Supernatants from in vitro cultured, sort-purified B cells were also collected for cytokine and antibody measurements. The concentrations of IgA, IgM, and IgG were determined using ELISA kits from eBioscience, following the manufacturer's protocols.

## B-cell purification and IgA class-switching assays

B cells were isolated from splenocytes using anti-B220 magnetic beads (Miltenyi Biotec) and cultured in triplicate in 24-well plates at a density of $5 \times 10^5$ cells per well. The purified B cells were incubated for 5 days in medium supplemented with αIgM (10 μg/ml), recombinant human TGF-β (2 ng/ml), anti-mCD40 (1 μg/ml), or BAFF (200 ng/ml). The frequency of B cells that underwent IgA class switching was determined by flow cytometry using FITC-conjugated anti-mouse IgA antibodies.

For in vivo IgA class-switching induction, mice were intraperitoneally immunized with 0.2 ml of sheep red blood cells (SRBC) ($1 \times 10^9$/ml in PBS). After 5 days, splenocytes and lymph node cells were analyzed via flow cytometry to detect IgA$^+$ B cells.

## B cell proliferation assay

Purified B cells were stimulated with various agents including anti-IgM (10 μg/ml), LPS (100 ng/ml), poly IC (20 μg/ml), R848 (1 μg/ml), anti-CD40 (1 μg/ml), or BAFF (200 ng/ml) in triplicate wells of 96-well plates, with $1 \times 10^5$ cells per well, at 37 °C. After 40 h of stimulation, B cell proliferation was assessed using the CCK8 assay.

## Immunofluorescence

Colons from 12-month-old mice were quickly frozen in Tissue-Tec OCT compound (VWR International) using liquid nitrogen-cooled 2-methylbutane. The frozen tissues were stored at $-70\,°C$ until sectioning into 2 μm cryostat sections. These sections were fixed in 100% acetone for 15 min and then stained overnight at 4 °C with fluorescein isothiocyanate (FITC)-labeled rat anti-mouse IgA or phycoerythrin (PE)-labeled rat anti-mouse IgM (BD Biosciences) at a concentration of 2 μg/ml.

## Immunoblot analysis

Purified B cells were stimulated with anti-IgM (10 μg/ml), anti-CD40 (1 μg/ml), or BAFF (500 ng/ml) for the specified durations. Total and subcellular extracts were then prepared from these cells and analyzed by immunoblotting (IB). For assessing protein phosphorylation, cells were lysed using a kinase cell lysis buffer that included phosphatase inhibitors.

## Overexpression in primary B cells

To achieve overexpression, retroviruses were generated by transfecting HEK293 cells with either an empty pCLXSN(GFP) vector or the same vector containing the genes of interest or their point mutants, along with the VSV-G and pCL-Ampho packaging vectors. Primary B cells were first stimulated with LPS (5 μg/ml) for 8 h, then infected with lentiviruses containing either the pCLXSN(GFP) vector or the vector with the targeted genes. After 72 h, the infected B cells were purified by flow cytometric sorting based on GFP expression and were subsequently used in in vitro IgA class-switching assays.

## NAT10 RIP–qPCR

B cells were lysed, with 10% of the lysate used as the "input" control and 80% subjected to immunoprecipitation using anti-NAT10 antibody, termed "IP." RNA was extracted from both the input and IP samples using TRIzol reagent (Invitrogen). The purified RNA was then eluted and analyzed by RT-qPCR.

## RNA stability assays

Cells were plated in 12-well plates at a density of $5 \times 10^4$ cells per well. On the second day, when cells reached approximately 40% confluence, doxycycline was added to the culture medium at a concentration of 1 μg/mL for 72 h. Following this, Actinomycin D (5 μg/mL) was introduced to inhibit mRNA transcription. Cells were collected after 6 and 12 h of Actinomycin D treatment, and total RNA was extracted using TRIzol reagent. Viral RNA levels were measured by RT-qPCR and analyzed using the $2^{-\Delta\Delta Ct}$ method, with residual RNAs normalized to the 0-h time point.

## Quantification and statistical analysis

Sample sizes were chosen according to standard guidelines and were also influenced by the availability of breeding resources. The number of independent samples in each group is represented by individual data points in the graphs and detailed in the figure legends. For behavioral experiments, the experimenters were blinded to the genotypes of the animals, and no samples were excluded from the analysis. Data are presented as mean ± SD. Statistical analyses were conducted using Prism 8.0 (GraphPad Software, La Jolla, CA, USA), with significance levels indicated as **$P < 0.01$ and ***$P < 0.005$. Comparisons between two groups were performed using a two-tailed unpaired $t$ test, while other comparisons were analyzed using one-way ANOVA. The specific statistical tests used for each analysis are noted in the figure legends. All data adhered to the assumptions required for these statistical tests, and the variances between groups were comparable.

## Data availability

The datasets produced in this study are available in the following databases: Genome Sequence Archive under the accession number PRJCA028331.

The source data of this paper are collected in the following database record: biostudies:S-SCDT-10_1038-S44319-025-00509-2.

## Peer review information

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

## Acknowledgements

This study was supported by distinguished Young Scientist Fund of NSFC (82125016), National Natural Science Foundation of China, Key Program (82230061), National Natural Science Foundation of China, "Pathogenesis and Intervention Strategies in Inflammatory Bowel Disease" Special projects (82341216). This research was supported by the Joint Funds of the Zhejiang Provincial Natural Science Foundation of China under Grant No. LHDMD22H100002 and LQ21H030013; This study was supported by the National Natural Science Foundation of China (82071046, 82100540) and the Natural Science Foundation of Jiangsu Province (Grant No. BK20211168). Project is also supported by the Young Scientists Fund of the National Natural Science Foundation of China (Grant No. 82202019). The project supported by the China Postdoctoral Science Foundation (Grant No. 2022M723664/ 2023M734007). This study was also supported by the 111 Program (D20036).

## Author contributions

**Wan-jun Jiang**: Formal analysis; Validation; Investigation; Visualization; Methodology; Writing—original draft. **Xin-tao Mao**: Resources; Data curation; Formal analysis; Validation; Investigation; Visualization; Methodology. **Wen-ping Li**: Conceptualization; Data curation; Validation; Investigation; Methodology. **Nicole Jin**: Data curation; Validation; Investigation; Methodology. **Yu Wang**: Data curation; Formal analysis; Validation; Investigation; Methodology. **Guiping Guan**: Supervision; Investigation; Methodology. **Jin Jin**: Supervision; Funding acquisition; Visualization; Writing—original draft; Writing—review and editing. **Yi-yuan Li**: Conceptualization; Resources; Supervision; Visualization; Writing—original draft; Project administration; Writing—review and editing.

Source data underlying figure panels in this paper may have individual authorship assigned. Where available, figure panel/source data authorship is listed in the following database record: biostudies:S-SCDT-10_1038-S44319-025-00509-2.

## Disclosure and competing interests statement

The authors declare no competing interests.

# Expanded View Figures

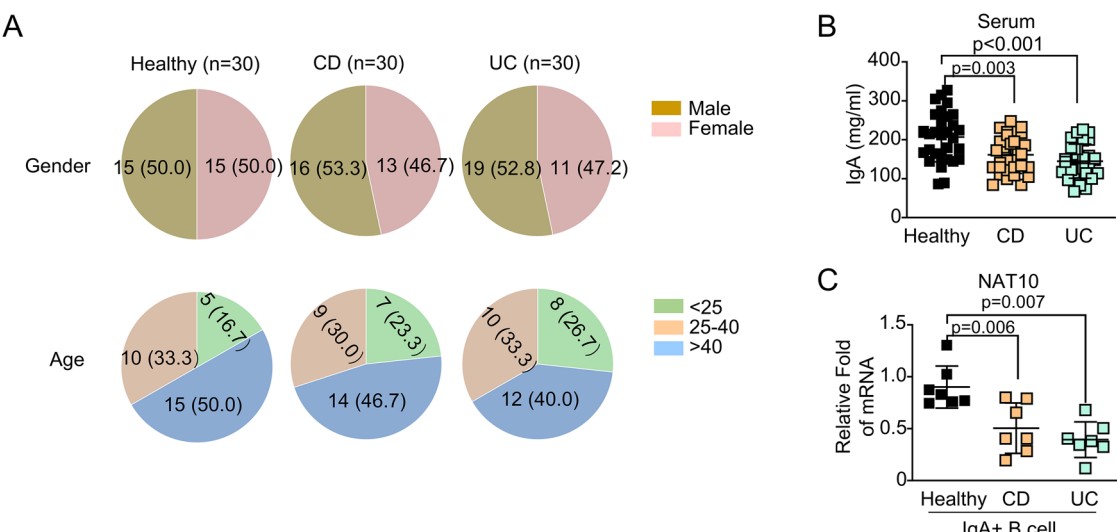

**Figure EV1.** Clinical information and grouping of IBD patients.

(A) Sex and age of patients with IBD and healthy volunteers ($n = 30$). (B) ELISA was used to quantify sIgA levels in serum from healthy donors and newly diagnosed patients with CD or UC ($n = 30$). (C) NAT10 expression was analyzed by qPCR in IgA+CD19+ B cells isolated from the colonic tissues of healthy donors and newly diagnosed CD or UC patients ($n = 7$). Results are shown as fold changes relative to *Actb* mRNA levels, normalized with Bio-Rad CFX Manager 3.1. All data are representative of biological replicates at least three independent experiments. Data are represented as the means ± SDs. The significance of differences (**B, C**) was determined using one-way ANOVA with Newman–Keuls post-hoc test. **P < 0.01; ***P < 0.005. Source data are available online for this figure.

A

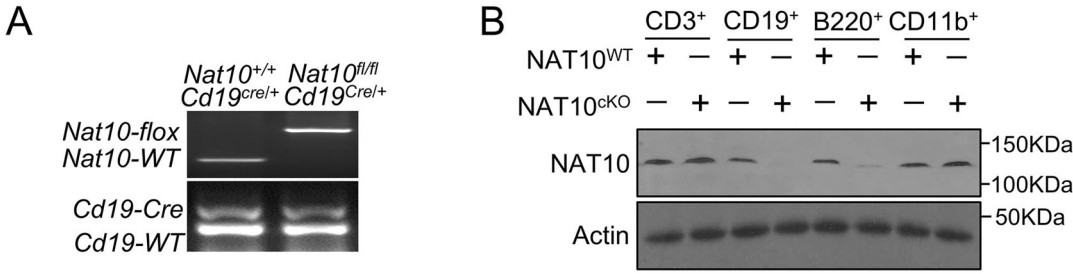

B

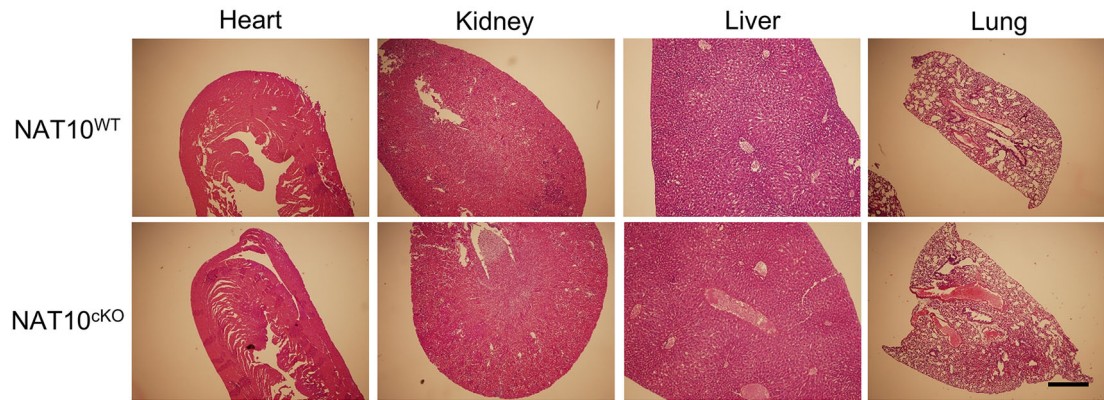

C

Figure EV2.    Validation of NAT10 knockout efficiency in B cells.

(A) NAT10cKO genotyping PCR. Nat10-floxed mice were crossed with CD19-Cre mice to generate *Nat10*fl/fl*Cd19*Cre/+ (NAT10cKO), and *Nat10*+/+*Cd19*Cre/+ (WT). (B) Immunoblotting (IB) assays showing specific ablation of NAT10 in the B cells of NAT10cKO mice. (C) Tissue sections of heart, kidney, liver, and lung from 6-week-old WT and NAT10cKO mice were performed with hematoxylin and eosin (H&E) staining (*n* = 3). All data are representative of biological replicates at least three independent experiments. Source data are available online for this figure.

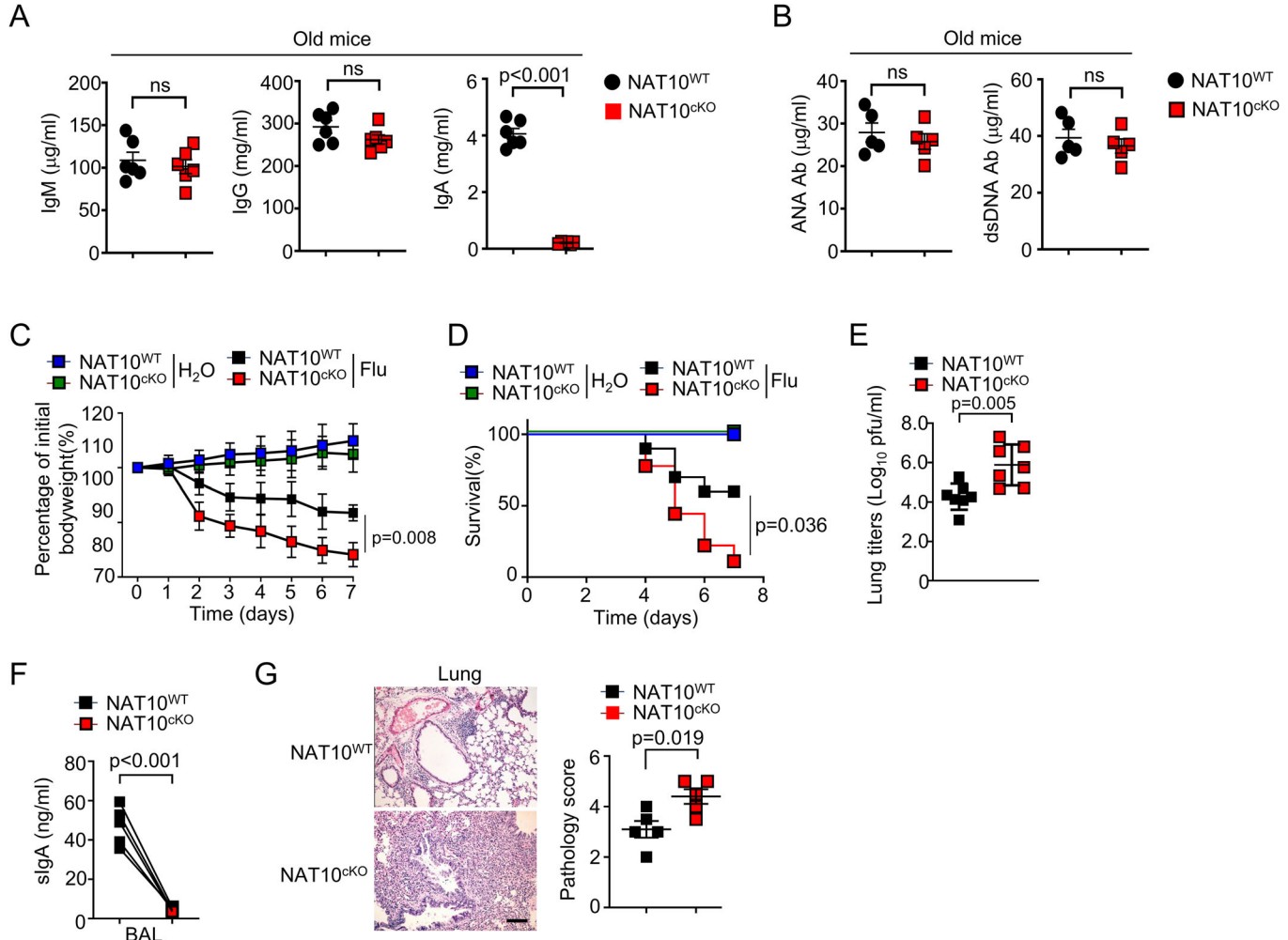

**Figure EV3. Assessment of IgA levels in aged mice.**

(A) Measurement of total antibody subclasses in the serum of 12-month-old WT and NAT10cKO mice using ELISA ($n = 6$). (B) ELISA analysis of baseline levels of autoantibodies against antinuclear antigen (ANA Ab) and double-stranded DNA (dsDNA Ab) in the serum of 12-month-old unimmunized WT and NAT10cKO mice ($n = 5$). (C) Body weight was tracked over a 7-day period in WT and NAT10cKO mice that were intranasally administered either influenza virus H7N9 or PBS ($n = 7$ per group). (D) Survival rates of H7N9-infected WT and NAT10cKO mice were monitored for 7 days ($n = 11$ per group). (E) Viral titers in the lungs were measured at 5 days post-infection (dpi) using a TCID50 assay ($n = 7$ per group). (F) ELISA was performed to assess IgA concentrations in the bronchoalveolar lavage fluid (BAL) at 5 dpi ($n = 5$). (G) Lung tissue sections from 6-week-old WT and NAT10cKO mice infected with H7N9 were stained with H&E for histological analysis ($n = 5$). All data are representative of biological replicates at least three independent experiments. Data are represented as the means ± SDs. The significance of differences (A, B) and (E, F) was determined by t test, and those (C, D) were determined using one-way ANOVA with Newman–Keuls post-hoc test. **$P < 0.01$; ***$P < 0.005$. Source data are available online for this figure.

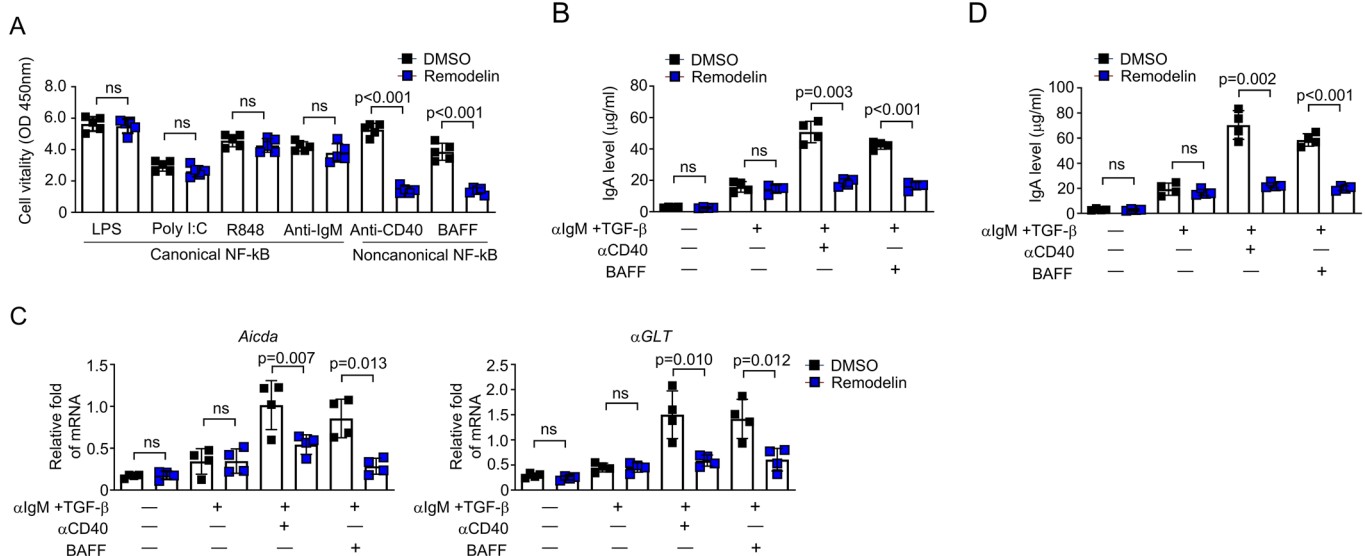

**Figure EV4. NAT10 inhibitor, Remodelin, enhances IgA production via noncanonical NF-κB pathway.**

(A) Proliferation assays of splenic B cells were incubated with complete medium containing 20 μM Remodelin, and stimulated with the specified inducers. These assays measured the proliferation of splenic B cells in response to the indicated stimuli ($n = 5$). (B) Intracellular IgA levels in splenic B cells were incubated with complete medium containing 20 μM Remodelin, and were analyzed after 5 days of culture with αIgM (10 μg/ml) alone or combined with TGF-β (2 ng/ml), anti-CD40 (α-CD40; 1 μg/ml), or BAFF (200 ng/ml). The percentage of IgA+ B cells was quantified by flow cytometry, with numbers shown in the outlined regions ($n = 4$). (C) qPCR was performed to assess α-GLT and AID mRNA expression in splenic B cells after 5 days of culture under the same conditions as in (B). The results are presented as fold changes relative to *Actb* mRNA levels and normalized using Bio-Rad CFX Manager 3.1 ($n = 4$). (D) ELISA was used to measure IgA levels in the supernatants of splenic B cells cultured for 5 days of culture under the same conditions as in (B) ($n = 4$). All data are representative of biological replicates at least three independent experiments. Data are represented as the means ± SDs. The significance of differences (A) was determined by $t$ test, and those (B, C) were determined using one-way ANOVA with Newman–Keuls post-hoc test. **$P < 0.01$; ***$P < 0.005$. Source data are available online for this figure.

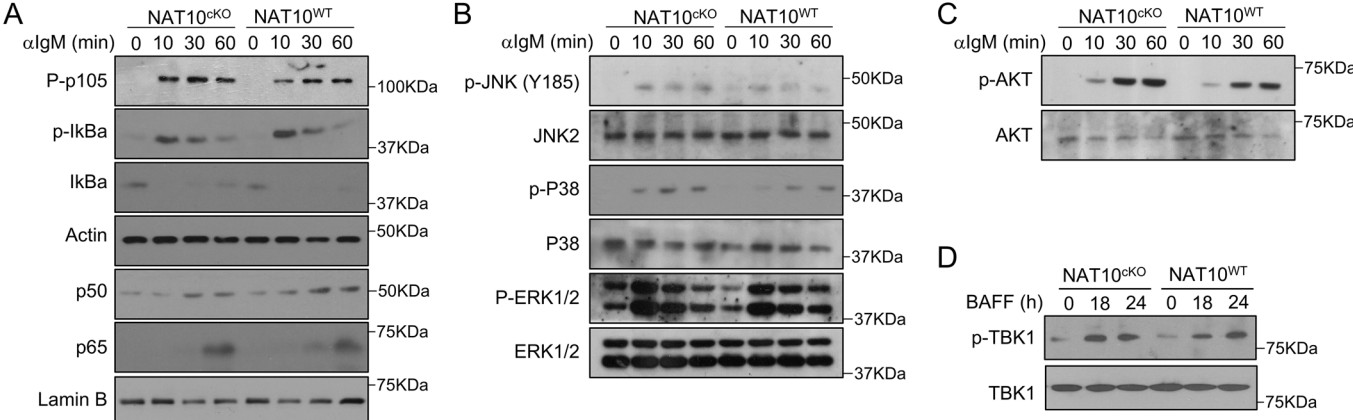

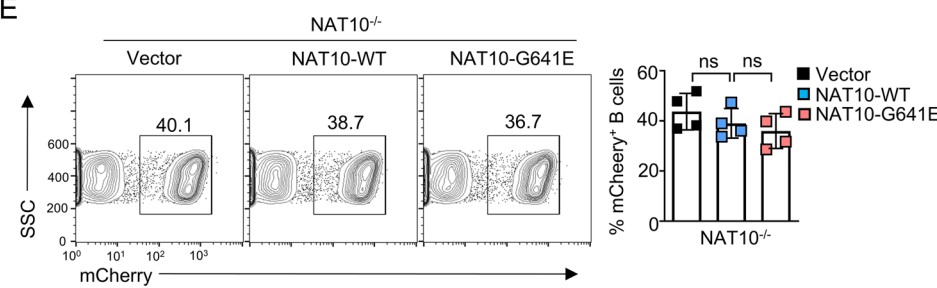

**Figure EV5.   NAT10 is dispensable for noncanonical NF-κB and MAPKs activation.**

(A–C) Spleen B cells derived from 6-weeks-old WT and NAT10cKO mice were stimulated as indicated. IB assays detecting the indicated NF-κB, MAPKs and AKT pathways and the loading control Lamin B in the nuclear extracts prepared from the αIgM stimulated WT and NAT10-deficient B cells. (D) IB analysis of TBK1 using total cell extracts from WT and NAT10-deficient B cells, stimulated as indicated. (E) Immunoassay of NAT10$^{-/-}$ B cells stimulated with LPS (5 μg/ml) for 8 h, followed by infection with a retroviral vector expressing GFP and either WT NAT10 (NAT10-WT) or the NAT10-G641E mutant. Infected cells (GFP$^+$) were analyzed by FACS ($n = 4$). All data are representative of biological replicates at least three independent experiments. Data are represented as the means ± SDs. The significance of differences (E) was determined using one-way ANOVA with Newman–Keuls post-hoc test. Source data are available online for this figure.

