## [Peer Review File · EMBO Reports]

NAT10-mediated acetylation of NIK mRNA in B cells promotes IgA production

Wan-jun Jiang, Xin-tao Mao, Wen-ping Li, Nicole Jin, Yu Wang, Guiping Guan, Jin Jin, and Yi-yuan Li

Corresponding author(s): Yi-yuan Li (103200067@seu.edu.cn), Jin Jin (jjin4@zju.edu.cn), Guiping Guan (guanguiping@hunau.edu.cn)

Review Timeline:

Submission Date:	23rd Jan 25
Editorial Decision:	18th Feb 25
Revision Received:	12th Mar 25
Editorial Decision:	8th Apr 25
Revision Received:	1st Jun 25
Accepted:	12th Jun 25

Editor: Achim Breiling

Transaction Report:

Dear Prof. Li,

Thank you for the submission of your manuscript to EMBO reports. I have now received the reports from the three referees that were asked to evaluate your study, which can be found at the end of this email.

As you will see, the referees think that these findings are of interest. However, they have several comments, concerns, and suggestions, indicating that a major revision of the manuscript is necessary to allow publication of the study in EMBO reports. As the reports are below, and all the referee concerns need to be addressed, I will not detail them here.

Given the constructive referee comments, I would like to invite you to revise your manuscript with the understanding that the concerns of the referees must be addressed in the revised manuscript and in a detailed point-by-point response. Acceptance of your manuscript will depend on a positive outcome of a second round of review. It is EMBO reports policy to allow a single round of revision only and acceptance of the manuscript will therefore depend on the completeness of your responses included in the next, final version of the manuscript.

- 1) a .docx formatted version of the final manuscript text (including legends for main figures, EV figures and tables), but without the figures included. Figure legends should be compiled at the end of the manuscript text.
- 2) individual production quality figure files as .eps, .tif, .jpg (one file per figure), of main figures and EV figures. Please upload these as separate, individual files upon re-submission.

- 4) a complete author checklist, which you can download from our author guidelines (<https://www.embopress.org/page/journal/14693178/authorguide>). Please insert page numbers in the checklist to indicate where the requested information can be found in the manuscript. The completed author checklist will also be part of the RPF.

- 5) that primary datasets produced in this study (e.g. RNA-seq, ChIP-seq, structural and array data) are deposited in an

appropriate public database. If no primary datasets have been deposited, please also state this in a dedicated section (e.g. 'No primary datasets have been generated and deposited'), see below.

The accession numbers and database should be listed in a formal "Data Availability" section that follows the model below. This is now mandatory (like the COI statement). Please note that the Data Availability Section is restricted to new primary data that are part of this study. This section is mandatory. As indicated above, if no primary datasets have been deposited, please state this in this section

Data availability

8) Regarding data quantification and statistics, please make sure that the number "n" for how many independent experiments were performed, their nature (biological versus technical replicates), the bars and error bars (e.g. SEM, SD) and the test used to calculate p-values is indicated in the respective figure legends (also for EV and Appendix figures). Please also check that all the p-values are explained in the legend, and that these fit to those shown in the figure. Please provide statistical testing where applicable. Please avoid the phrase 'independent experiment', but clearly state if these were biological or technical replicates. Please also indicate (e.g. with n.s.) if testing was performed, but the differences are not significant. In case n=2, please show the data as separate datapoints without error bars and statistics. See also: <http://www.embopress.org/page/journal/14693178/authorguide#statisticalanalysis>

9) Please add scale bars of similar style and thickness to microscopic images, using clearly visible black or white bars (depending on the background). Please place these in the lower right corner of the images themselves. Please do not write on or near the bars in the image but define the size in the respective figure legend.

10) Please also note our reference format:

12) We now use CRediT to specify the contributions of each author in the journal submission system. CRediT replaces the author contribution section. Please use the free text box to provide more detailed descriptions and do NOT provide your final manuscript text file with an author contributions section. See also our guide to authors: <https://www.embopress.org/page/journal/14693178/authorguide#authorshipguidelines>

13) All Materials and Methods need to be described in the main text using our 'Structured Methods' format, which is required for

all research articles. According to this format, the Methods section should include a Reagents and Tools Table (listing key reagents, experimental models, software, and relevant equipment and including their sources and relevant identifiers), uploaded as separate file, and a Methods section in which we encourage the authors to describe their methods using a step-by-step protocol format with bullet points, to facilitate the adoption of the methodologies across labs. More information on how to adhere to this format as well as downloadable templates (.doc) for the Reagents and Tools Table can be found in our author guidelines (section 'Structured Methods'):

14) Please order the sections like this, using these names:

Title page - Abstract - Keywords - Introduction - Results - Discussion - Methods - Data availability section - Acknowledgements (including the funding information) - Disclosure and Competing Interests Statement - References - Figure legends - Expanded View Figure legends

15) Please make sure that all the funding information is also entered into the online submission system and that it is complete and similar to the one in the acknowledgement section of the manuscript text file.

Please note that corresponding authors are required to supply an ORCID ID upon submission of a revised manuscript and an institutional e-mail address. Please do for Jin Jin and Guiping Guan. Please find instructions on how to link the ORCID ID to the account in our manuscript tracking system in our Author guidelines:

<http://www.embopress.org/page/journal/14693178/authorguide#authorshipguidelines>

I look forward to seeing a revised form of your manuscript when it is ready.

Yours sincerely,

Referee #1:

Jiang et al. are the first to demonstrate the crucial role of the N-acetyltransferase NAT10 as a regulator of noncanonical NF- κ B-mediated IgA⁺ B-cell development. They uncover a novel function of NAT10 in acetylating and stabilizing NIK mRNA in B cells, which ultimately enhances NIK protein expression and activates the NF- κ B2 pathway. The authors investigated the impact of NAT10 deficiency on serum IgA titers in immunized mice following exposure to both T-dependent and T-independent antigen complexes. They also explored the severity of gastrointestinal disorders associated with NAT10KO-mediated IgA deficiency.

The main originality of the study lies in the discovery of NAT10 as a crucial regulator of IgA B-cell development. The manuscript is well-written and the data are solid. However, minor comments are listed below:

- The mechanistic understanding of the critical role of NAT10 in IgA class-switching is missing.

- The authors show a significant reduction in NAT10 expression in colonic B cells from IBD patients. They also suggest that NAT10 may play a role in the regulation of IgA production by B cell in IBD. However, the authors should further substantiate this claim with additional discussion given the reduced frequency of IgA⁺ B cells already documented in the IBD patient's vs healthy individuals.

If NAT10 plays a role in regulating IgA production in IBD, one would expect differences in its expression levels between IgA⁺ B cells from IBD patients and healthy individuals.

- Accordingly, the authors should discuss the significant reduction in NAT10 expression in colonic B cells from IBD patients. How NAT10 is negatively regulated in IBD? It would be interesting to explore whether NAT10 expression is reduced in chronic inflammatory conditions like in IBD.

Has NAT10 overexpression been attempted in B cells in vivo? Testing this hypothesis in DSS-induced colitis models could provide insights into NAT10's therapeutic potential.

Referee #2:

The authors did a good job of capturing the impact of NAT10 in IgA+ B cells and how NAT10 absence contributes to pathology of IBD through loss of NAT10-ac4C dependent NIK RNA stability. However, authors need to address the following:

- In figure 1C, change the arrangement of igA- and igA+ to correspond with your arrangement for western blot.
- After line 176-177, authors should state a conclusive insight from there results as to why NAT10 is dispensable for B cell development and maturation.
- In figure 3 (B, C and E), authors should be consistent arrangement of panels i.e 1B is IgA, IgG and IgM; and 1C, is IgM, IgG and IgA.
- Authors should check typos like line 231should be NAT10-regulated and not NAT1-regulated.
- Typo in line 279---- B cells/ When.
- In figure 6G, it is noticeable that the protein of G641E is not expressed when compared to vector, do you think this could be the reason why there isn't recovery of IgA+ B cells as illustrated in 6F. Why is the protein unstable, perhaps authors should explain why G641E is unstable using structural models because what these results is showing is the effect of NAT10's G641E on igA+ B cells is due to structural instability and not ac4C-dependent.
- Type in figure 7F, should be mouse instead of mucus.
- Authors showed predicted ac4C site in CDS region of Map3k14(NIK) catalyzed by NAT10 in figure 7E, it would be better to isolated RNA, treated with NaCNBH3 or NaBH4, PCR amplify this region and do a sanger seq in NAT10WT and NAT10cKO spleen B cells or at least HEK293T cells.

Referee #3:

In this manuscript by Jian et al., the authors show the role of N-acetyltransferase NAT10 in regulating IgA production by B cells through acetylation of Map3k14 (NIK). By analyzing patient samples, genetically that engineered mice and cell line models, they demonstrate NAT10 level correlates with IgA production. Further, they show evidence of NIK mRNA acetylation by NAT10 strengthens noncanonical NF- B pathway as a novel regulator of IgA class-switch recombination. The data in the manuscript supports well their conclusions, even though the extent of NAT10 mediated acetylation in CSR remains to be further explored. I have some concerns about this manuscript which should be addressed before publication.

1. In Figure1, the authors should define CD and UC groups in the figure legend.
2. NAT10 mRNA level CD19+ B cells from healthy individuals in Figure 1E shows half the amount compared to Actb , while it is >30 fold over Actb in healthy controls from colonic tissues (Figure 1G). This seems in consistent, since the respective protein levels in Figures 1F,1H do not correspond to mRNA data. The authors need to explain the large variations.
3. In Figure 5, splenic B cells viabilities were shown to be reduced in response to non-canonical NF- B pathway inducers in NAT10 KO cells. I am curious to know if NAT10 levels are altered in the WT B cells in response to canonical and noncanonical NF- B inducers. In response to CD40 stimulation, WT B cells seem to be upregulating NAT10 protein level (Figure 6C). This will strengthen the argument of NAT10 being upstream or downstream of NF- B pathways.
4. The authors should use consistency in displaying qPCR results. Why is the qPCR analysis in Figure 7A is displayed differently than other places in the manuscript? The normalization, errors and statistical significance of these data is not known. How are the data in 7G normalized?
5. How does NAT10 affect TBK1 levels? Since TBK1 has been shown to negatively regulate NIK (Jin et al., Nature Immunology, 2012) and NAT10 is shown positive regulator, do they affect each other?
6. Since AID levels were reduced NAT10KO, do the authors know why IgG class-switching is not affected?
7. Please add Figure numbers. It is not easy to navigate the Figures without Figure numbers.

Referee #1:

Jiang et al. are the first to demonstrate the crucial role of the N-acetyltransferase NAT10 as a regulator of noncanonical NF- κ B-mediated IgA⁺ B-cell development. They uncover a novel function of NAT10 in acetylating and stabilizing NIK mRNA in B cells, which ultimately enhances NIK protein expression and activates the NF- κ B2 pathway. The authors investigated the impact of NAT10 deficiency on serum IgA titers in immunized mice following exposure to both T-dependent and T-independent antigen complexes. They also explored the severity of gastrointestinal disorders associated with NAT10KO-mediated IgA deficiency. The main originality of the study lies in the discovery of NAT10 as a crucial regulator of IgA B-cell development. The manuscript is well-written and the data are solid. However, minor comments are listed below:

1. The mechanistic understanding of the critical role of NAT10 in IgA class-switching is missing.

Response: We are sorry for not clearly describing our conclusion. We sincerely appreciate the reviewer's insightful comment regarding the mechanistic role of NAT10 in IgA class-switching. As highlighted in our study (Figure 7), we identified that NAT10 catalyzes N4-acetylcytidine (ac4C) modifications on NIK mRNA, thereby enhancing its stability and translational efficiency. This acetylation-dependent stabilization of NIK mRNA promotes sustained activation of the noncanonical NF- κ B pathway, which is indispensable for AID expression and IgA CSR. CSR to different isotypes exhibits varying sensitivities to AID levels. Studies suggest IgG CSR requires lower AID activity compared to IgA.

1. IgA CSR is primarily driven by TGF- β signaling, which induces activation of Smad proteins and the *Germline α (GL α)* promoter³. This pathway may require higher thresholds of AID expression or stability to initiate recombination. IgA CSR is primarily driven by TGF- β signaling, which induces activation of Smad proteins and the *Germline α (GL α)* promoter. This pathway may require higher thresholds of AID expression or stability to initiate recombination^{1,2,3}.

2. IgG CSR (e.g., IgG1, IgG2a) is regulated by cytokines like IL-4 or IFN- γ , which activate STAT6 or STAT1, respectively. These pathways might tolerate lower AID levels due to compensatory mechanisms, such as enhanced transcriptional activity of *Germline γ (GL γ)* promoters. IgG CSR (e.g., IgG1, IgG2a) is regulated by cytokines like IL-4 or IFN- γ , which activate STAT6 or STAT1, respectively. These pathways might tolerate lower AID levels due to compensatory mechanisms, such as enhanced transcriptional activity of *Germline γ (GL γ)* promoters⁴.

These studies suggested that AID dosage effects are context-dependent, with IgA CSR being more sensitive to AID depletion. In this manuscript, we emphasize the mechanism to clarify

how NAT10 bridges RNA acetylation to NF- κ B-dependent IgA regulation.

2. The authors show a significant reduction in NAT10 expression in colonic B cells from IBD patients. They also suggest that NAT10 may play a role in the regulation of IgA production by B cell in IBD. However, the authors should further substantiate this claim with additional discussion given the reduced frequency of IgA⁺ B cells already documented in the IBD patient's vs healthy individuals. If NAT10 plays a role in regulating IgA production in IBD, one would expect differences in its expression levels between IgA⁺ B cells from IBD patients and healthy individuals.

Response: The reviewers presented a novel mechanistic hypothesis positing that diminished NAT10 expression may underlie the depletion of intestinal IgA⁺ B cells observed in IBD pathogenesis. While BAFF and CD40L signaling pathways are recognized as collaborative regulators of IgA⁺ plasma cell maintenance, their functional impairment in IBD – driven by inflammatory microenvironmental perturbations – likely exacerbates B cell apoptotic susceptibility and compromises mucosal IgA synthesis. Notably, our investigations revealed that NAT10 expression was transcriptionally regulated by non-canonical NF- κ B pathway activators, including BAFF and CD40L. This regulatory nexus suggested a potential divergence in NAT10 expression profiles between healthy individuals and IBD patients at the

IgA⁺ B cell level. To address this, we performed flow cytometry-assisted isolation of lamina propria-derived IgA⁺ B cells from matched cohorts. qPCR analyses demonstrated a significant downregulation of NAT10 transcripts in IBD-derived IgA⁺ B cells compared to healthy controls (**Supplementary Figure 1C**). These findings corroborate the reviewer-proposed model wherein NAT10 deficiency disrupts IgA⁺ B cell survival or differentiation.

3. Accordingly, the authors should discuss the significant reduction in NAT10 expression in colonic B cells from IBD patients. How NAT10 is negatively regulated in IBD? It would be interesting to explore whether NAT10 expression is reduced in chronic inflammatory conditions like in IBD. Has NAT10 overexpression been attempted in B cells in vivo? Testing this hypothesis in DSS-induced colitis models could provide insights into NAT10's therapeutic potential.

Response: Firstly, we sincerely thank the reviewer for raising this important question. As the reviewer rightly pointed out, the negative regulatory mechanism of NAT10 expression in IBD is indeed a crucial scientific issue. While as mentioned in the previous response, we speculate that the phenomenon may be due to decreased expression or impaired function of proteins such as BAFF or CD40L that promote NAT10 expression. Insufficient BAFF/CD40L signaling might reduce NIK mRNA production, creating a negative feedback loop that further diminishes NAT10 activity. Additionally, we propose that the following mechanisms may also regulate NAT10 expression:

1. Inflammatory Cytokines: Pro-inflammatory cytokines such as TNF- α , IL-6, or IL-1 β , which are elevated in IBD, may suppress NAT10 expression. These cytokines could activate signaling pathways (e.g., NF- κ B or STAT3) that directly or indirectly downregulate NAT10 transcription.
2. Microbial Influence: The characteristic dysbiosis in IBD could alter microbial metabolites (e.g., short-chain fatty acids) or pathogen-associated molecular patterns (e.g., LPS), which might impair NAT10 expression via Toll-like receptor (TLR) signaling.

We have added the above discussions to the Discussion section.

Furthermore, while our current study employs a knockout model without examining NAT10 overexpression, we acknowledge that NAT10 overexpression in B cells could provide in vivo validation of its functional role. However, due to the limited delivery efficiency and targeting specificity of viral vectors or nanoparticle-based systems in vivo, we believe our in vitro findings sufficiently support our conclusions.

Referee #2:

The authors did a good job of capturing the impact of NAT10 in IgA+ B cells and how NAT10 absence contributes to pathology of IBD through loss of NAT10-ac4C dependent NIK RNA

stability. However, authors need to address the following:

1. In figure 1C, change the arrangement of igA- and igA+ to correspond with your arrangement for western blot.

Response: We thank the reviewer's nice remind. We changed the arrangement of igA- and igA+ as suggestion in Figure 1C.

2. After line 176-177, authors should state a conclusive insight from there results as to why NAT10 is dispensable for B cell development and maturation.

Response: According the reviewer's suggestion, we included a statement as following: "As known, B cell development relies heavily on NF- κ B signaling for survival and differentiation, as highlighted in studies focusing on NF- κ B's role in activated B cell-like diffuse large B cell lymphoma and normal B cell homeostasis^{5,6}. These pathways operate independently of NAT10-mediated RNA acetylation or cell-cycle regulation, which are critical in other cellular contexts (e.g., oocyte maturation or cytokinesis)."

3. In figure 3 (B, C and E), authors should be consistent arrangement of panels i.e 1B is IgA, IgG and IgM; and 1C, is IgM, IgG and IgA.

Response: The reviewer's suggestion was well-taken. We adjusted the order of presentation of IgA, IgG, and IgM.

4. Authors should check typos like line 231should be NAT10-regulated and not NAT1-regulated.

Response: We thank the reviewer's nice remind. We corrected this typo.

5. Typo in line 279---- B cells/ When.

Response: We thank the reviewer's nice remind. We corrected this typo.

6. In figure 6G, it is noticeable that the protein of G641E is not expressed when compared to vector, do you think this could be the reason why there isn't recovery of IgA+ B cells as illustrated in 6F. Why is the protein unstable, perhaps authors should explain why G641E is unstable using structural models because what these results is showing is the effect of NAT10's G641E on igA+ B cells is due to structural instability and not ac4C-dependent.

Response: We appreciate the reviewer's question and apologize for any insufficient clarity in describing our experimental results. As demonstrated in Figure 6G, the NAT10 mutant protein

is indeed expressed. However, the post-stimulation NIK protein levels in cells expressing the NAT10 mutant remain comparable to those observed in the NAT10 KO-vector group, indicating a failure to effectively induce NIK protein accumulation. Previous studies have established that the G641E mutation in NAT10 significantly impairs its acetyltransferase activity (citation). Our findings thus suggest that the impact of the NAT10 G641E mutant on IgA⁺ B cells underscores the critical role of its RNA ac4C modification activity in this regulatory mechanism.

7. Type in figure 7F, should be mouse instead of mucus.

Response: We thank the reviewer's nice remind. We corrected this typo.

8. Authors showed predicted ac4C site in CDS region of Map3k14(NIK) catalyzed by NAT10 in figure 7E, it would be better to isolated RNA, treated with NaCNBH3 or NaBH4, PCR amplify this region and do a sanger seq in NAT10WT and NAT10cKO spleen B cells or at least HEK293T cells.

Response: We sincerely thank the reviewer for their insightful suggestion to directly validate the ac4C sites in NIK mRNA via chemical stabilization and Sanger sequencing. While this approach would indeed strengthen our mechanistic claims, we respectfully note that our current experimental evidence already provides robust support for NAT10-mediated ac4C modification of NIK mRNA:

1. The NIKac4C mut exhibited significantly reduced mRNA stability and failed to rescue IgA production in NIK-deficient B cells (Figure 7G-J), directly linking ac4C modification to NIK functionality.
2. Prior studies (e.g., Wang et al., Clin Transl Med 2022; Yan et al., Nat Commun 2023) have established that NAT10-dependent ac4C modifications predominantly localize to coding sequences (CDS) to stabilize target mRNAs, aligning with our bioinformatic predictions (Figure 7F).

Thank you for highlighting this point, which underscores the importance of ac4C in post-transcriptional regulation. We believe our existing data robustly support the proposed mechanism, but we remain open to further validation as resources permit.

Referee #3:

In this manuscript by Jian et al., the authors show the role of N-acetyltransferase NAT10 in regulating IgA production by B cells through acetylation of Map3k14 (NIK). By analyzing patient samples, genetically that engineered mice and cell line models, they demonstrate NAT10 level correlates with IgA production. Further, they show evidence of NIK mRNA acetylation by NAT10 strengthens noncanonical NF- κ B pathway as a novel regulator of IgA

class-switch recombination. The data in the manuscript supports well their conclusions, even though the extent of NAT10 mediated acetylation in CSR remains to be further explored. I have some concerns about this manuscript which should be addressed before publication.

1. In Figure 1, the authors should define CD and UC groups in the figure legend.

Response: As suggested by the reviewer, we have supplemented the definitions of these two abbreviations.

2. NAT10 mRNA level CD19+ B cells from healthy individuals in Figure 1E shows half the amount compared to Actb, while it is >30 fold over Actb in healthy controls from colonic tissues (Figure 1G). This seems inconsistent, since the respective protein levels in Figures 1F, 1H do not correspond to mRNA data. The authors need to explain the large variations.

Response: We are grateful to the reviewer for identifying an error in the labeling of Figure 1G, which has been corrected in the revised manuscript. Regarding the quantitative PCR (qPCR) data analysis, all gene expression levels were normalized to the endogenous control β -actin (Actb) mRNA using the comparative Ct ($\Delta\Delta C_t$) method. Data normalization and statistical analysis were performed using Bio-Rad CFX Manager software (version 3.1), which implements an automated normalization algorithm. This algorithm centers the mean expression values between experimental groups at 1.0-fold change to facilitate comparative analysis and interpretation of relative gene expression patterns.

3. In Figure 5, splenic B cell viabilities were shown to be reduced in response to non-canonical NF- κ B pathway inducers in NAT10 KO cells. I am curious to know if NAT10 levels are altered in the WT B cells in response to canonical and noncanonical NF- κ B inducers. In response to α CD40 stimulation, WT B cells seem to be upregulating NAT10 protein level (Figure 6C). This will strengthen the argument of NAT10 being upstream or downstream of NF- κ B pathways.

Response: This represents a pivotal question in our study. We have demonstrated that inducers of the non-canonical NF- κ B pathway, including BAFF and anti-CD40, significantly upregulate NAT10 expression (**Figure 1E-F** and **Figure 6C**). These findings support a mechanistic model wherein the non-canonical NF- κ B pathway and NAT10 engage in a positive feedback loop. This reciprocal interaction not only amplifies non-canonical NF- κ B signaling activity but also drives the differentiation of B cells into IgA⁺ populations.

4. The authors should use consistency in displaying qPCR results. Why is the qPCR analysis in Figure 7A displayed differently than other places in the manuscript? The normalization,

errors and statistical significance of these data is not known. How are the data in 7G normalized?

Response: We sincerely appreciate the reviewer's insightful comments regarding the presentation of qPCR data. The heatmap visualization employed in Figure 7A was strategically selected to facilitate comparative analysis of multiple non-canonical NF- κ B pathway-associated gene expression profiles across experimental conditions. This graphical approach enables simultaneous visualization of relative expression patterns for multiple targets while maintaining intergroup comparability - particularly advantageous when analyzing parallel transcriptional responses within complex signaling networks. The experimental methodology section has been augmented to clarify that all qPCR results were normalized against β -actin (Actb) mRNA levels using the $\Delta\Delta$ Ct method. Data normalization and statistical analysis were performed utilizing Bio-Rad CFX Manager 3.1 software (v3.1.1517.0823), which implements automatic baseline correction and cycle threshold determination algorithms. The software's default normalization protocol centers group means at unity (1.0-fold change) to optimize visual interpretation of relative expression differences while maintaining mathematical integrity of the dataset.

5. How does NAT10 affect TBK1 levels? Since TBK1 has been shown to negatively regulate NIK (Jin et al., Nature Immunology, 2012) and NAT10 is shown positive regulator, do they affect each other?

Response: We thank the reviewer for raising this critical point. As noted, prior seminal work by co-authors of this study demonstrated that BAFF promotes TBK1 phosphorylation and induces NIK phosphorylation, thereby destabilizing NIK. To evaluate whether NAT10 influences this axis, we compared TBK1 phosphorylation levels in BAFF-stimulated WT and NAT10 KO B cells. Our data revealed no significant difference in TBK1 activation between these groups (**Supplementary Figure 6D**), thus demonstrating that NAT10-mediated regulation of the non-canonical NF- κ B pathway operates independently of TBK1 activity.

6. Since AID levels were reduced NAT10KO, do the authors know why IgG class-switching is not affected?

Response: This is an excellent question. The observed dissociation between IgA and IgG class-switching defects in NAT10^{CKO} cells, despite reduced AID expression may reflect distinct regulatory requirements for different antibody isotypes during class-switch recombination (CSR). CSR to different isotypes exhibits varying sensitivities to AID levels. Studies suggest IgG CSR requires lower AID activity compared to IgA.

1. IgA CSR is primarily driven by TGF- β signaling, which induces activation of Smad proteins and the *Germline α* (*GL α*) promoter³. This pathway may require higher thresholds of

AID expression or stability to initiate recombination. IgA CSR is primarily driven by TGF- β signaling, which induces activation of Smad proteins and the *Germline α* (*GL α*) promoter. This pathway may require higher thresholds of AID expression or stability to initiate recombination^{1,2,3}.

2. IgG CSR (e.g., IgG1, IgG2a) is regulated by cytokines like IL-4 or IFN- γ , which activate STAT6 or STAT1, respectively. These pathways might tolerate lower AID levels due to compensatory mechanisms, such as enhanced transcriptional activity of *Germline γ* (*GL γ*) promoters. IgG CSR (e.g., IgG1, IgG2a) is regulated by cytokines like IL-4 or IFN- γ , which activate STAT6 or STAT1, respectively. These pathways might tolerate lower AID levels due to compensatory mechanisms, such as enhanced transcriptional activity of *Germline γ* (*GL γ*) promoters⁴.

These studies suggested that AID dosage effects are context-dependent, with IgA CSR being more sensitive to AID depletion.

7. Please add Figure numbers. It is not easy to navigate the Figures without Figure numbers.

Response: The reviewer's suggestion was well-taken.

Reference

1. Cerutti A. The regulation of IgA class switching. *Nature Reviews Immunology* 2008, **8**(6): 421-434.
2. Jang YS, Seo GY, Lee JM, Seo HY, Han HJ, Kim SJ, *et al.* Lactoferrin causes IgA and IgG2b isotype switching through betaglycan binding and activation of canonical TGF- β signaling. *Mucosal Immunology* 2015, **8**(4): 906-917.
3. Lee J-M, Jang Y-S, Jin B-R, Kim S-J, Kim H-J, Kwon B-E, *et al.* Retinoic acid enhances lactoferrin-induced IgA responses by increasing betaglycan expression. *Cellular & Molecular Immunology* 2016, **13**(6): 862-870.

4. Mikocziova I, Greiff V, Sollid LM. Immunoglobulin germline gene variation and its impact on human disease. *Genes & Immunity* 2021, **22**(4): 205-217.

5. Zhao M, Chauhan P, Sherman CA, Singh A, Kaileh M, Mazan-Mamczarz K, *et al.* NF- κ B subunits direct kinetically distinct transcriptional cascades in antigen receptor-activated B cells. *Nature Immunology* 2023, **24**(9): 1552-1564.

6. Sasaki Y, Iwai K. Roles of the NF- κ B Pathway in B-Lymphocyte Biology. In: Kurosaki T, Wienands J (eds). *B Cell Receptor Signaling*. Springer International Publishing: Cham, 2016, pp 177-209.

Dear Prof. Li,

Thank you for the submission of your revised manuscript to our editorial offices. I have now received the reports from the three referees that were asked to re-evaluate the study, you will find below. As you will see, the referees now support its publication in EMBO reports. However, referee #2 has remaining concerns and suggestions to improve the manuscript, I ask you to address in a final revised manuscript. Please do the requested changes to the manuscript. Please also provide a final p-b-p-response regarding these points.

- I suggest this revised title:

NAT10-mediated acetylation of NIK mRNA promotes IgA production of B cells

- Please provide the abstract written in present tense throughout.

- Please note that corresponding authors are required to supply an ORCID ID upon submission of a revised manuscript and an institutional e-mail address. Please do this for co-corresponding authors Jin and Guan. Please find instructions on how to link the ORCID ID to the account in our manuscript tracking system in our Author guidelines:
<http://www.embopress.org/page/journal/14693178/authorguide#authorshipguidelines>

Please also provide institutional e-mail addresses for all the corresponding authors on the title page of the manuscript and in the submission system.

- Please provide individual production quality figure files as .eps, .tif, .jpg (one file per figure), of the EV figures (presently named Supplementary Figures). Please upload these - as the main figures - as separate, individual files upon re-submission and name these Figure EVx in their filename, the legends and all the callouts. Please provide their legend at the end of the manuscript text file (see also below).

- For all figures the size and resolution is very low, resulting e.g. in pixelated Western blots. Please provide high resolution figure files (main and EV figures) for your final revised manuscript.

- The presently uploaded document 'Supplementary Figure legends' lists 6 figures, but only 5 Supplementary Figures have been provided. Please check.

- Please order the manuscript sections like this, using these names:

Title page - Abstract - Keywords - Introduction - Results - Discussion - Methods - Data availability section - Acknowledgements (including the funding information) - Disclosure and Competing Interests Statement - References - Figure legends - Expanded View Figure legends

- Please move the ethics paragraph to the Methods section.

- Please remove the mention of source data from the Data availability section.

- Please use our reference format (with 'et al' used for citations with more than 10 author names):

- Please provide a complete author checklist, filling in also boxes D66 and D69, and regarding data availability (it seems datasets have been deposited for this study).

- Please check again that the number "n" for how many independent experiments were performed, their nature (biological versus technical replicates), the bars and error bars (e.g. SEM, SD) and the test used to calculate p-values is indicated in the respective figure legends. Please also check that all the p-values are explained in the legend, and that these fit to those shown in the figure. Please provide statistical testing where applicable. Please avoid the phrase 'independent experiment', but clearly state if these were biological or technical replicates. Please also indicate (e.g. with n.s.) if testing was performed, but the differences are not significant. In case n=2, please show the data as separate datapoints without error bars and statistics. See also:

<http://www.embopress.org/page/journal/14693178/authorguide#statisticalanalysis>

If n<5, please show single datapoints for diagrams. Moreover:

- Please note that the exact p values are not provided in the legends of figures 1A, B, C, E, G; 3B-E; 4A, B, F, G, H, I, J; 5A-C; 6E, F; 7B, D, E, G, I, J; supplementary figures 1B, C; 3A, E, F; 4A-D

- Please note that in figures 2A-D; 3B-E there is a mismatch between the annotated p values in the figure legend and the

annotated p values in the figure file that should be corrected.

- Please note that information related to n is missing in the legends of figures 1E, supplementary figure 2C.
- Please note that the scale bar needs to be defined for figure 2E, supplementary figure 3G
- Please add scale bars of similar style and thickness to microscopic images, using clearly visible black or white bars (depending on the background). Please place these in the lower right corner of the images themselves. Please do not write on or near the bars in the image but define the size in the respective figure legend. Presently, some scale bars are too thin or have text nearby. Please check.
- Please move the primer information (Table S1) to the 'Reagents & Tools Table' and update any callouts.
- Please make sure that all the funding information is also entered into the online submission system and that it is complete and similar to the one in the acknowledgement section of the manuscript text file. Presently, grant from "Pathogenesis and Intervention Strategies in Inflammatory Bowel Disease" Special projects (82341216); Joint Funds of the Zhejiang Provincial Natural Science Foundation of China under Grant No. LHDMD22H100002 and LQ21H030013; the Natural Science Foundation of Jiangsu Province (Grants No BK20211168); Young Scientists Fund of the National Natural Science Foundation of China (Grant No. 82202019); the China Postdoctoral Science Foundation (Grant No. 2022M723664/ 2023M734007); the 111 Program (D20036)" are missing in the submission system. Please check.
- Please upload the source data as one folder per figure, grouping together all the files for this figure figure (and ZIPed together), and as one folder for the EV figures.
- The source data file for Fig. 7E (attached) contains several duplicated values (marked in blue, orange or red), which seems unlikely for a qPCR experiment. This might be caused by the methodology used, but please check and comment on this in your final p-b-p-response.
- Please also have your final manuscript carefully proofread by a native speaker. There are still grammatical errors present.

In addition, I would need from you uploaded separately:

Best,
Achim Breiling
Senior Editor
EMBO Reports

Referee #1:

The authors answered clearly my questions, and improve substantially the manuscript in clarity.

I have no further questions

Referee #2:

1. In figure 1C, change the arrangement of igA- and igA+ to correspond with your arrangement for western blot.
Response: We thank the reviewer's nice remind. We changed the arrangement of igA- and igA+ as suggestion in Figure 1C.
Reviewer's comment: The Comment is addressed by the authors.
2. After line 176-177, authors should state a conclusive insight from there results as to why NAT10 is dispensable for B cell development and maturation.

Response: According to the reviewer's suggestion, we included a statement as follows:

"As known, B cell development relies heavily on NF- κ B signaling for survival and differentiation, as highlighted in studies focusing on NF- κ B's role in activated B cell-like diffuse large B cell lymphoma and normal B cell homeostasis^{5, 6}. These pathways operate independently of NAT10-mediated RNA acetylation or cell-cycle regulation, which are critical in other cellular contexts (e.g., oocyte maturation or cytokinesis)."

Reviewer's comment: The concern addressed is not satisfactory. (i) Authors should always reply, especially when including text in the manuscript, with line numbers for easy tracking. (ii) Authors made state that "NF- κ B signaling pathway is independent of NAT10-mediated RNA acetylation..." however, articles (PMID: 38058058, PMID: 37307924; PMID: 37644005) claim that NAT10 promotes/regulates NF- κ B pathways. Perhaps authors should have a close look at the statement and address as appropriate. I strongly suggest authors to address this concern.

3. In figure 3 (B, C and E), authors should be consistent arrangement of panels i.e 1B is IgA, IgG and IgM; and 1C, is IgM, IgG and IgA.

Response: The reviewer's suggestion was well-taken. We adjusted the order of presentation of IgA, IgG, and IgM.

Reviewer's comment: The concern addressed by authors is satisfactory.

4. Authors should check typos like line 231should be NAT10-regulated and not NAT1-regulated.

Response: We thank the reviewer's nice remind. We corrected this typo.

Reviewer's comment: Satisfactory

5. Typo in line 279---- B cells/ When.

Response: We thank the reviewer's nice remind. We corrected this typo.

Reviewer's comment: Satisfactory

6. In figure 6G, it is noticeable that the protein of G641E is not expressed when compared to vector, do you think this could be the reason why there isn't recovery of IgA+ B cells as illustrated in 6F. Why is the protein unstable, perhaps authors should explain why G641E is unstable using structural models because what these results is showing is the effect of NAT10's G641E on IgA+ B cells is due to structural instability and not ac4C-dependent.

Response: We appreciate the reviewer's question and apologize for any insufficient clarity in describing our experimental results. As demonstrated in Figure 6G, the NAT10 mutant protein is indeed expressed. However, the post-stimulation NIK protein levels in cells expressing the NAT10 mutant remain comparable to those observed in the NAT10 KO-vector group, indicating a failure to effectively induce NIK protein accumulation. Previous studies have established that the G641E mutation in NAT10 significantly impairs its acetyltransferase activity (citation). Our findings thus suggest that the impact of the NAT10 G641E mutant on IgA+ B cells underscores the critical role of its RNA ac4C modification activity in this regulatory mechanism.

Reviewer's comment: Noted. I hope the authors will give this mutation a closer look in future projects, as unstable proteins or poorly expressed proteins can lead to a biased interpretation of the results. Otherwise, I have no further comments here.

7. Typo in figure 7F, should be mouse instead of mucus.

Response: We thank the reviewer's nice remind. We corrected this typo.

Reviewer's comment: Satisfactory

8. Authors showed predicted ac4C site in CDS region of Map3k14(NIK) catalyzed by NAT10 in figure 7E, it would be better to isolate RNA, treated with NaCNBH3 or NaBH4, PCR amplify this region and do a sanger seq in NAT10WT and NAT10cKO spleen B cells or at least HEK293T cells.

Response: We sincerely thank the reviewer for their insightful suggestion to directly validate the ac4C sites in NIK mRNA via chemical stabilization and Sanger sequencing. While this approach would indeed strengthen our mechanistic claims, we respectfully note that our current experimental evidence already provides robust support for NAT10-mediated ac4C modification of NIK mRNA:

1. The NIKac4C mutant exhibited significantly reduced mRNA stability and failed to rescue IgA production in NIK-deficient B cells (Figure 7G-J), directly linking ac4C modification to NIK functionality.

2. Prior studies (e.g., Wang et al., Clin Transl Med 2022; Yan et al., Nat Commun 2023) have established that NAT10-dependent ac4C modifications predominantly localize to coding sequences (CDS) to stabilize target mRNAs, aligning with our bioinformatic predictions (Figure 7F).

Thank you for highlighting this point, which underscores the importance of ac4C in post-transcriptional regulation. We believe our existing data robustly support the proposed mechanism, but we remain open to further validation as resources permit.

Reviewer's comment: While the authors fail to address the concern here. I advise the authors to apply an orthogonal approach in assessing ac4C modification, i.e., base resolution sequencing method and Sanger sequencing, to assess ac4C etc. Predictors and RIP-qPCR could yield false positive results. I have no further comment on this concern.

Referee #3:

The authors have sufficiently addressed my concerns/suggestions on the manuscript. I recommend its publication in EMBO reports.

1. I suggest this revised title: *NAT10-mediated acetylation of NIK mRNA promotes IgA production of B cells*

Response: We corrected this typo as suggestion.

2. Please provide the abstract written in present tense throughout.

Response: We revised our abstract as suggestion.

- Please note that corresponding authors are required to supply an ORCID ID upon submission of a revised manuscript and an institutional e-mail address. Please do this for co-corresponding authors Jin and Guan. Please find instructions on how to link the ORCID ID to the account in our manuscript tracking system in our Author guidelines: <http://www.embopress.org/page/journal/14693178/authorguide#authorshipguidelines>

Response: As editors' suggestion, we provided these information.

Please also provide institutional e-mail addresses for all the corresponding authors on the title page of the manuscript and in the submission system.

Response: As editors' suggestion, we provided these information.

- Please provide individual production quality figure files as .eps, .tif, .jpg (one file per figure), of the EV figures (presently named Supplementary Figures). Please upload these - as the main figures - as separate, individual files upon re-submission and name these Figure EVx in their filename, the legends and all the callouts. Please provide their legend at the end of the manuscript text file (see also below).

Response: As editors' suggestion, we provided these information.

- For all figures the size and resolution is very low, resulting e.g. in pixelated Western blots. Please provide high resolution figure files (main and EV figures) for your final revised manuscript.

Response: As editors' suggestion, we provided the high-resolution images.

- The presently uploaded document 'Supplementary Figure legends' lists 6 figures, but only 5 Supplementary Figures have been provided. Please check.

Response: We sincerely apologize for the overlapping text that occurred during the conversion of the revised document, which led to the misinterpretation of six Supplementary Figure legends. We have now corrected this error.

- Please order the manuscript sections like this, using these names:
Title page - Abstract - Keywords - Introduction - Results - Discussion - Methods - Data availability section - Acknowledgements (including the funding information) - Disclosure and Competing Interests Statement - References - Figure legends - Expanded View Figure legends

Response: Thank you for your guidance. I have reordered the manuscript sections as requested

- Please move the ethics paragraph to the Methods section.

Response: Thank you for your further guidance. I have moved the ethics paragraph to the Methods section as requested.

- Please remove the mention of source data from the Data availability section.

Response: We revised the manuscript as suggestion.

- Please use our reference format (with 'et al' used for citations with more than 10 author names):

Response: We revised the manuscript as suggestion.

- Please provide a complete author checklist, filling in also boxes D66 and D69, and regarding data availability (it seems datasets have been deposited for this study).

Response: We revised the author checklist as suggestion.

- Please check again that the number "n" for how many independent experiments were performed, their nature (biological versus technical replicates), the bars and error bars (e.g. SEM, SD) and the test used to calculate p-values is indicated in the respective figure legends. Please also check that all the p-values are explained in the legend, and that these fit to those shown in the figure. Please provide statistical testing where applicable. Please avoid the phrase 'independent experiment', but clearly state if these were biological or technical replicates. Please also indicate (e.g. with n.s.) if testing was performed, but the differences are not significant. In case n=2, please show the data as separate datapoints without error bars and statistics. See also:

<http://www.embopress.org/page/journal/14693178/authorguide#statisticalanalysis>

Response: Thank you for your feedback. We have carefully revised all figure legends to address the statistical reporting requirements. The revisions include:

Clarification of replicates: All *n* values now explicitly state the number of biological replicates unless otherwise noted. Technical replicates are not used for statistical comparisons.

Error bars: All data are presented as mean \pm SD.

Avoidance of vague terms: Phrases like “independent experiments” have been replaced with “biological replicates.”

If n<5, please show single datapoints for diagrams. Moreover:

- Please note that the exact p values are not provided in the legends of figures 1A, B, C, E, G; 3B-E; 4A, B, F, G, H, I, J; 5A-C; 6E, F; 7B, D, E, G, I, J; supplementary figures 1B, C; 3A, E, F; 4A-D

- Please note that in figures 2A-D; 3B-E there is a mismatch between the annotated p values in the figure legend and the annotated p values in the figure file that should be corrected.

- Please note that information related to n is missing in the legends of figures 1E, supplementary figure 2C.

- Please note that the scale bar needs to be defined for figure 2E, supplementary figure 3G

Response: We made the revision as editors' suggestion.

- Please add scale bars of similar style and thickness to microscopic images, using clearly visible black or white bars (depending on the background). Please place these in the lower right corner of the images themselves. Please do not write on or near the bars in the image but define the size in the respective figure legend. Presently, some scale bars are too thin or have text nearby. Please

check.

Response: We made the revision as editors' suggestion.

- Please move the primer information (Table S1) to the 'Reagents & Tools Table' and update any callouts.

Response: We combined Table S1 to the Reagents & Tools Table as suggestion.

- Please make sure that all the funding information is also entered into the online submission system and that it is complete and similar to the one in the acknowledgement section of the manuscript text file. Presently, grant from "Pathogenesis and Intervention Strategies in Inflammatory Bowel Disease" Special projects (82341216); Joint Funds of the Zhejiang Provincial Natural Science Foundation of China under Grant No. LHDMD22H100002 and LQ21H030013; the Natural Science Foundation of Jiangsu Province (Grants No BK20211168); Young Scientists Fund of the National Natural Science Foundation of China (Grant No. 82202019); the China Postdoctoral Science Foundation (Grant No. 2022M723664/2023M734007); the 111 Program (D20036)" are missing in the submission system. Please check.

Response: As editors' suggestion, we provided these information.

- Please upload the source data as one folder per figure, grouping together all the files for this figure figure (and ZIPed together), and as one folder for the EV figures.

Response: We uploaded the files as suggestion.

- The source data file for Fig. 7E (attached) contains several duplicated values (marked in blue, orange or red), which seems unlikely for a qPCR experiment. This might be caused by the methodology used, but please check and comment on this in your final p-b-p-response.

Response: The editors were right. The excessive number of decimal places in the NAT10cKO group's data was caused by the use of a different output mode. We have reprocessed and re-exported these data, but the original data figures remain unchanged.

- Please also have your final manuscript carefully proofread by a native speaker. There are still grammatical errors present.

Response: As requested, we have carefully proofread the manuscript with a focus on grammatical accuracy and language fluency.

In addition, I would need from you uploaded separately:

- a short, two-sentence summary of the manuscript (not more than 35 words).

- two to four short (!) bullet points highlighting the key findings of your study (two lines each).

- a schematic summary figure as separate file that provides a sketch of the major findings (not a data image) in jpeg or tiff format (with the exact width of 550 pixels and a height of not more than 400 pixels) that can be used as a visual synopsis on our website.

Response: We provided these necessary document and image as suggestion.

Referee #2:

2. After line 176-177, authors should state a conclusive insight from there results as to why NAT10

is dispensable for B cell development and maturation.

Response: According to the reviewer's suggestion, we included a statement as following:

"As known, B cell development relies heavily on NF- κ B signaling for survival and differentiation, as highlighted in studies focusing on NF- κ B's role in activated B cell-like diffuse large B cell lymphoma and normal B cell homeostasis^{5, 6}. These pathways operate independently of NAT10-mediated RNA acetylation or cell-cycle regulation, which are critical in other cellular contexts (e.g., oocyte maturation or cytokinesis)."

Reviewer's comment: The Concern addressed is not satisfactory. (i) Authors should always reply, especially when including text in the manuscript, with line numbers for easy tracking. (ii) Authors made state that "NF- κ B signaling pathway is independent of NAT10-mediated RNA acetylation..." however, articles (PMID: 38058058, PMID: 37307924; PMID: 37644005) claim that NAT10 promotes/regulates NF- κ B pathways, Perhaps authors should have a close look at the statement and address as appropriate. I strongly suggest authors to address this concern.

Response: We corrected this statement as reviewer's suggestion.

6. In figure 6G, it is noticeable that the protein of G641E is not expressed when compared to vector, do you think this could be the reason why there isn't recovery of IgA+ B cells as illustrated in 6F. Why is the protein unstable, perhaps authors should explain why G641E is unstable using structural models because what these results is showing is the effect of NAT10's G641E on igA+ B cells is due to structural instability and not ac4C-dependent.

Response: We appreciate the reviewer's question and apologize for any insufficient clarity in describing our experimental results. As demonstrated in Figure 6G, the NAT10 mutant protein is indeed expressed. However, the post-stimulation NIK protein levels in cells expressing the NAT10 mutant remain comparable to those observed in the NAT10 KO-vector group, indicating a failure to effectively induce NIK protein accumulation. Previous studies have established that the G641E mutation in NAT10 significantly impairs its acetyltransferase activity (citation). Our findings thus suggest that the impact of the NAT10 G641E mutant on IgA+ B cells underscores the critical role of its RNA ac4C modification activity in this regulatory mechanism.

Reviewer's comment: Noted. I hope the authors will give this mutation a closer look in future projects, as unstable proteins or poorly expressed proteins can lead to a biased interpretation of the results. Otherwise, I have no further comments here.

Response: We will fully take the suggestion from the reviewers.

Prof. Yi-yuan Li
Southeast University
Key Laboratory for Developmental Genes and Human Disease, Ministry of Education, Institute of Life Sciences, Jiangsu
Province High-Tech Key Laboratory for Bio-Medical Research, Southeast University
Nanjing, Jiangsu 210096
China

Dear Prof. Li,

I am very pleased to accept your manuscript for publication in the next available issue of EMBO reports. Thank you for your contribution to our journal.

Yours sincerely,
